# Differential whole-genome doubling and homologous recombination deficiencies across breast cancer subtypes from the Taiwanese population

Chia-Hsin Wu [1,16], Chia-Shan Hsieh[2,16], Yo-Cheng Chang[3,16], Chi-Cheng Huang[4], Hsien-Tang Yeh[5], Ming-Feng Hou [6], Yuan-Chiang Chung[7], Shih-Hsin Tu[8], King-Jen Chang[9], Amrita Chattopadhyay[10], Liang-Chuan Lai [11], Tzu-Pin Lu [12], Yung-Hua Li[1], Mong-Hsun Tsai [13✉] & Eric Y. Chuang [1,10,14,15✉]

Whole-genome doubling (WGD) is an early macro-evolutionary event in tumorigenesis, involving the doubling of an entire chromosome complement. However, its impact on breast cancer subtypes remains unclear. Here, we performed a comprehensive and quantitative analysis of WGD and its influence on breast cancer subtypes in patients from Taiwan and consequently highlight the genomic association between WGD and homologous recombination deficiency (HRD). A higher manifestation of WGD was reported in triple-negative breast cancer, conferring high chromosomal instability (CIN), while HER2 + tumors exhibited early WGD events, with widely varied CIN levels, compared to luminal-type tumors. An association of higher activity of de novo indel signature 2 with WGD and HRD in Taiwanese breast cancer patients was reported. A control test between WGD and pseudo non-WGD samples was further employed to support this finding. The study provides a better comprehension of tumorigenesis in breast cancer subtypes, thus assisting in personalized treatment.

[1] Graduate Institute of Biomedical Electronics and Bioinformatics, National Taiwan University, Taipei, Taiwan. [2] Genome and Systems Biology Degree Program, National Taiwan University, Taipei, Taiwan. [3] Yonglin Healthcare Foundation, New Taipei City, Taiwan. [4] Comprehensive Breast Health Center, Taipei Veterans General Hospital, Taipei, Taiwan. [5] Department of Surgery, Lotung Poh-Ai Hospital, Yilan County, Taiwan. [6] Division of Breast Surgery, Department of Surgery, Kaohsiung Medical University, Kaohsiung, Taiwan. [7] Department of Breast Surgery, Dajia Branch, Kuang Tien General Hospital, Taichung, Taiwan. [8] Department of Surgery, School of Medicine, College of Medicine, Taipei Medical University, Taipei, Taiwan. [9] Department of Surgery, National Taiwan University Hospital, Taipei, Taiwan. [10] Bioinformatics and Biostatistics Core, Centers of Genomic and Precision Medicine, National Taiwan University, Taipei, Taiwan. [11] Graduate Institute of Physiology, College of Medicine, National Taiwan University, Taipei, Taiwan. [12] Department of Public Health, National Taiwan University, Taipei, Taiwan. [13] Institute of Biotechnology, National Taiwan University, Taipei, Taiwan. [14] Department of Electrical Engineering, National Taiwan University, Taipei, Taiwan. [15] Master Program for Biomedical Engineering, China Medical University, Taichung, Taiwan. [16]These authors contributed equally: Chia-Hsin Wu, Chia-Shan Hsieh, Yo-Cheng Chang. ✉email: motiont@gmail.com; chuangey@ntu.edu.tw

Breast cancer is the most commonly diagnosed cancer and the leading cause of cancer-related death worldwide[1]. In recent years, there has been a sharp rise in breast cancer incidence in the Asia-Pacific region, which has brought increased visibility of Asians as a distinct breast cancer patient population comprising 22.8% of all cases globally[1]. Notably, the incidence of breast cancer in the Asia-Pacific population exhibits remarkable variation (range of age-standardized rate: 32.8–59.8 per 100,000 population)[1]. Taken together, thorough genomic characterization of the Asian cohort will provide valuable insights for studying cancer genomics and cancer heterogeneity.

Most molecular studies of breast cancer, which have contributed greatly toward identifying clinical subtype-specific genes, have been based on single-nucleotide variants (SNVs) and copy number alterations (CNAs)[2]. Such genetic variations offer a limited view of underlying breast cancer etiology as breast cancer is a multifactorial disease and variations such as these fail to provide a complete picture, and therefore, must be complemented by large-scale chromosomal abnormalities, which dominate the genomic landscape of cancer. In this study, the previous biological finding of breast cancer subtypes caused by different profiles of SNVs and CNAs[2] allows us to hypothesize that large-scale somatic events might also explain the subtype heterogeneity. Whole-genome doubling (WGD), the doubling of a complete set of diploid chromosomes, has been proposed as an early event in tumor evolution, resulting in tetraploidy[3,4]. WGD induces chromosomal instability (CIN) and is associated with poor prognosis in many cancers[3,4]. It is believed to originate from errors in cell cycle progression due to a defective G1 checkpoint[5]. However, the extent of WGD and CIN within subtypes of breast cancer has not been explored yet in the Asian cohort.

Furthermore, despite the fact that WGD is a punctuated event in tumor evolution, its interplay with other large-scale events, such as dysfunction of DNA repair processes, remains unclear. Homologous recombination, a conservative error-free mechanism to repair double-strand breaks (DSBs), operates during the late S and G2 phases of the cell cycle[6]. Homologous recombination deficiency (HRD) leads to an alternative error-prone DSB repair process known as nonhomologous end-joining, yielding characteristic genomic deletions[6,7]. In breast cancer, HRD is a common genomic feature, which predicts response to DNA-damaging agents[8,9]. Although recent efforts have reported associations between HRD and bi-allelic loss-of-function of homologous recombination-related genes BRCA1/2[10], the involved mechanisms are still not clearly defined. Furthermore, the extent of homologous recombination deficiency in the WGD samples hasn't been quantitatively checked in the Asian cohort before.

To address such lack of information, we performed a comprehensive analysis of deep whole-exome sequencing (WES) data from breast cancer patients in the Taiwanese population (BCTW), to infer the impact of WGD, along with frequent cancer gene alterations, on the timing of events driving tumor initiation and tumor maintenance within subtypes of breast cancer (hormone receptor-positive and human epidermal growth factor 2 receptor-negative (HR + /HER2-), HR + /HER2 + , HER2 + , and triple-negative (TNBC)). WGD and HRD were revealed to co-occur with the specific indel pattern, where further genomic characterization of alternative homologous recombination repair processes revealed indel signatures to be a better and more reliable predictor of WGD-linked HRD phenotype than the substitution signatures. This study aspires to shed light on the understanding of tumorigenesis by revealing the molecular basis for breast cancer subtypes, with the goal of enhancing personalized treatment strategies.

## Results

**Patient characteristics**. A total of 116 patients were diagnosed between 27 and 85 years (median 53.0 years) of age, frequently at early stages (80% at stage I-II) and with an intermediate histologic grade (53%). The majority of BCTW patients (60%) were classified with luminal A subtype (defined as HR + /HER2-), while the remaining patients were classified evenly among HER2 + (15%), TNBC (13%) and luminal B (HR + /HER2 + ; 12%) subtypes. Table 1 summarizes the clinical characteristics of all patients included in this study.

**Mutational landscape of BCTW samples**. A total of 13,174 somatic SNVs (median 53.5), including 5375 nonsynonymous variants (median 26) and 1374 somatic small insertions/deletions (indels; median 5), were identified from the patients in this study. The median tumor mutation burden (TMB) was 1 mutation/Mb; however, a few breast cancer patients demonstrated hyper-mutation with a TMB of more than 10 mutations per Mb. (Fig. 1a). Interestingly, TMB was reported to be positively associated with the mutational status of gene PIK3CA (P = 0.034) using a one-tailed Mann–Whitney U test.

To detect the significantly mutated genes (SMGs) in BCTWs, four robust approaches (Supplementary Fig. 1) were applied to genes with expression in human breast tissue (genes with no expression were excluded)[11]. The results revealed the following: (1) the frequency-based approach (dNdScv) identified the gene COMP as a BCTW-specific SMG that has not been included previously in the Cancer Gene Census-v86[12,13] (Supplementary Note 1). Another four genes, TP53, PIK3CA, GATA3, and AKT1

### Table 1 Patient and Sample Characteristics.

| Characteristics | Patients (N = 116)* |
|---|---|
| Age, years (n = 115) | |
| Median (interquartile range) | 53.0 (45.5–64.5) |
| Distribution | |
| ≤40 | 15 (13) |
| 41–50 | 40 (35) |
| 51–60 | 20 (17) |
| 61–70 | 23 (20) |
| >70 | 17 (15) |
| Pathologic stage (n = 112) | |
| I | 33 (29) |
| II | 57 (51) |
| III | 19 (17) |
| IV | 3 (3) |
| Histologic grade of tumor (n = 113) | |
| Low | 17 (15) |
| Intermediate | 60 (53) |
| High | 36 (32) |
| Estrogen receptor expression (n = 114) | |
| Negative | 32 (28) |
| Positive | 82 (72) |
| Progesterone receptor expression (n = 114) | |
| Negative | 51 (45) |
| Positive | 63 (55) |
| Human epidermal growth factor 2 receptor expression | |
| Negative | 78 (73) |
| Positive | 29 (27) |
| Immunohistochemical classification (n = 107) | |
| HR + /HER2- | 64 (60) |
| HR + /HER2+ | 13 (12) |
| HER2+ | 16 (15) |
| TNBC | 14 (13) |

HR hormone receptor, HER2 human epidermal growth factor 2 receptor, TNBC triple-negative breast cancer.
*All values are presented as n (%) unless otherwise indicated.

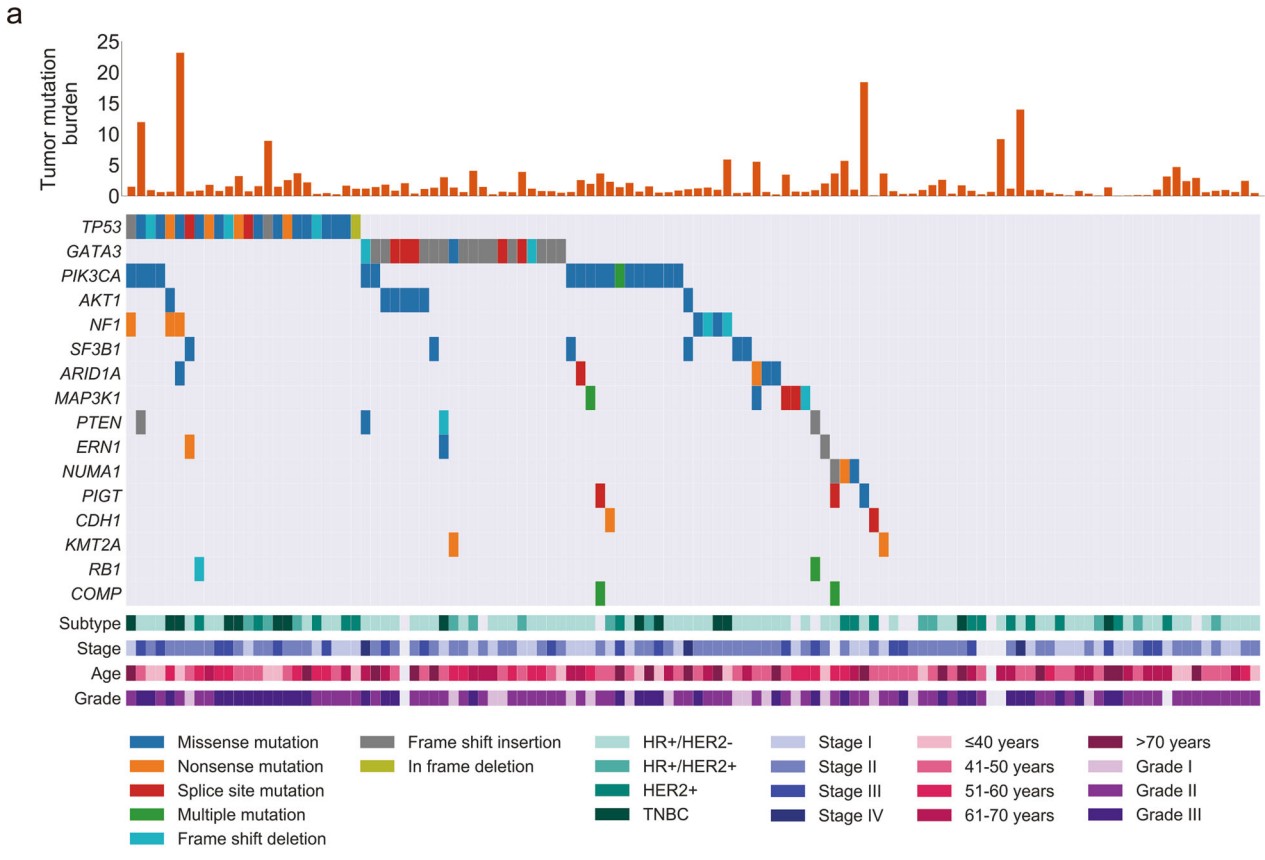

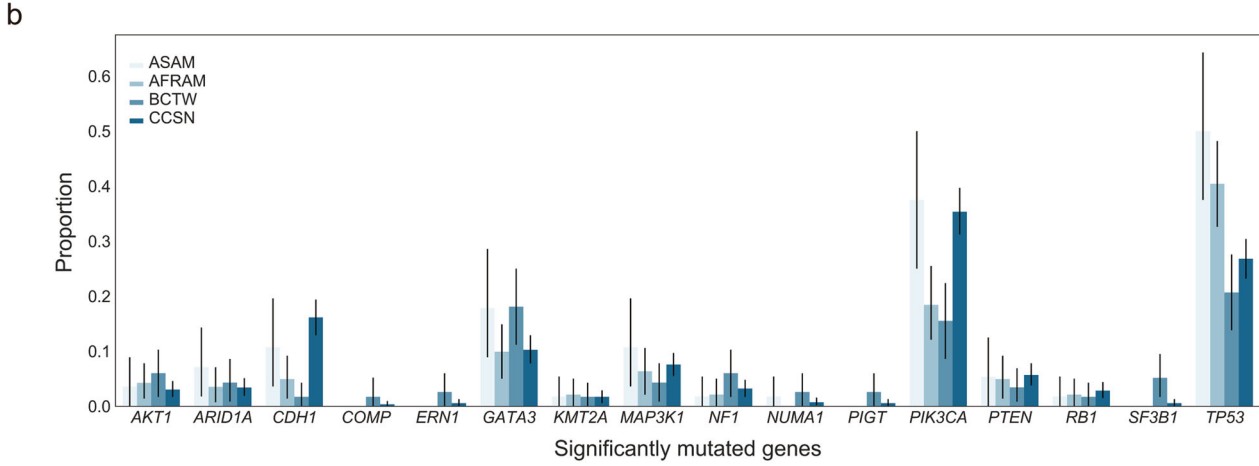

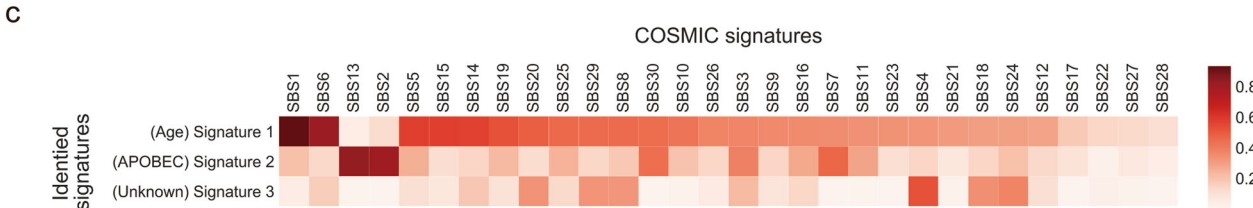

(previously reported), were also reported as SMGs under positive selection, with a higher prevalence of positively accumulated mutations than expected[14]. Somatic mutations in *PIK3CA* were predominantly missense, while *TP53* exhibited a variety of alterations (Fig. 1a). (2) To complement the aforementioned frequency-based approach, the 20/20 rule-based approach[15], which evaluates the proportion of missense mutations and loss-of-function mutations in the gene of interest, identified 10 additional SMGs, including *SF3B1*, *NF1*, *MAP3K1*, *ARID1A*, *NUMA1*, *CDH1*, *RB1*, *KMT2A*, *PTEN*, and a BCTW-specific SMG, *ERN1*. (3) Assessment of mutations with high functional impact (OncodriveFML, a domain-based approach)[16] revealed

**Fig. 1 Mutational landscape of somatic alteration and single base substitution mutational signatures (SBSs) in 116 breast cancer samples. a** Rows represent significantly mutated genes (SMGs), and columns represent individual tumors. Samples are arranged to emphasize mutual exclusivity among alterations. SMGs are ordered according to the frequency of nonsynonymous single-nucleotide variations/indels. The stacked bar plot depicts the tumor mutation burden (TMB; mutations/covered bases; y-axis) for individual tumors (x-axis). Key clinical features are annotated for each tumor. Clinical characteristics and mutation types are indicated with color. **b** The bar plot with the 95% confidence interval indicates mutational frequencies of BCTWs, compared with those from The Cancer Genome Atlas benchmark cohorts (right; BCTW, Taiwanese; CCSN, Caucasian; AFRAM, African American; ASAM, Asian American). **c** Heatmap of the cosine similarity results for the three de novo SBSs of the Taiwanese population (y-axis), coded by color. In the scale bar, the cosine similarity (range 0–1) represents the extent of similarity to a particular signature of COSMIC (x-axis). Among the 30 COSMIC SBSs, the APOBEC- and age-related signatures were the most similar mutational signatures detected in the Taiwanese population (dark red), while one signature was dissimilar to any COSMIC mutational signatures and thus is considered unknown.

the well-known SMGs *GATA3* and *TP53*. (4) Parsing locally clustered 'hotspot mutations' (OncodriveCLUST, the feature-based approach)[17] identified 1 BCTW-specific SMG, *PIGT*, occurring in 2.6% of BCTW samples.

The identified SMGs were then compared across breast cancer subtypes in BCTWs. *TP53* remained largely unaltered in the HR + /HER- subtype (14.1%), but *TP53* mutations were enriched in both HR + /HER2 + and HER2 + cancers (23.1%, 33.3%) (Fig. 1a), with the highest frequency of *TP53* mutations (50.0%) in TNBC tumors. The prevalence of *PIK3CA* mutations was similar between the HR + /HER2- (17.2%) and TNBC (21.4%) subtypes and lower in HER2 + tumors (6.7%) than in HR + /HER2 + tumors (15.4%).

Comparative analysis across populations was briefly performed using BCTWs and The Cancer Genome Atlas (TCGA) benchmark cohorts (Fig. 1b). The prevalence of *GATA3* mutation in BCTWs was the same as that in Asian Americans. Genes *KMT2A* and *RB1* displayed similar mutation prevalence for both the BCTW and TCGA cohorts, except for Caucasians. Interestingly, two of the most commonly mutated genes in BCTWs, *TP53* and *PIK3CA* (20.7 and 15.5%, respectively), were present at lower frequencies in BCTWs than in TCGA cohorts (26.8 and 35.4% in Caucasians; 40.4 and 18.4% in African Americans; 50.0 and 37.5% in Asian Americans). Higher mutation frequency in *CDH1* and *MAP3K1* were also observed in TCGA cohorts as compared to BCTWs. In contrast, the BCTW cohort had higher mutation rates in *AKT1*, *NF1*, and *SF3B1*. Our analysis also identified somatic mutations in genes that turned out to be only locally prevalent (*ERN1*, *PIGT*, and *COMP*) (Fig. 1b).

To characterize the mutational processes that contributed to the BCTW cohort, a mutational signature analysis was performed. Among the 30 identified pan-cancer single base substitution (SBS) mutational signatures released in the Catalogue of Somatic Mutations in Cancer (COSMIC)[18], there are 13 SBS signatures related to breast cancer[19]. A cosine similarity analysis, to compare the de novo SBS mutational signatures (de novo SBS) derived from BCTW samples with those from COSMIC, identified two signatures that were highly similar to COSMIC SBS signatures (COSMIC SBS; Fig. 1c). De novo SBS1 was highly similar to an age-dependent COSMIC SBS1[19] (cosine similarity 0.933), while de novo SBS2 displayed high similarity to COSMIC SBS2 and COSMIC SBS13 (cosine similarity 0.806 and 0.834, respectively), which might be due to overactivity of the APOBEC family of cytidine deaminases[19]. However, de novo SBS3 did not show much similarity to any of the COSMIC SBS signatures (all cosine similarity <0.55).

**WGD and CIN in breast cancer subtypes**. In total, 19.0% of BCTW patients were found with tumors that underwent WGD. The rate of WGD exhibited remarkable variation across breast cancer subtypes, affecting 50.0% of TNBC samples versus only 14.1% of HR + /HER2- tumors, 23.1% of HR + /HER2 + and 20.0% of HER2 + tumors (Fig. 2a). The TMB of WGD-positive

tumors was found to be higher than those of WGD-negative tumors (median TMB of 2.018 and 0.835, respectively; $P = 1.484 \times 10^{-6}$; one-tailed Mann–Whitney $U$ test). Next, compared to non-WGD samples, the relatively frequent alterations of cancer genes were found in WGD samples, where genes *MYC*, *EIF4EBP1*, and *FGFR1* displayed amplifications; *DUSP4*, *LEPROTL1*, *NRG1*, and *WRN* deletions; and *MUC16* demonstrated mutations ($P < 0.05$, odds ratio >1; logistic regression). HR + /HER2 + tumors, further, exhibited a greater degree of CIN than HR + /HER- tumors, while the highest level of CIN was observed in TNBC samples, consistent with their higher WGD frequency (Fig. 2b). Additionally, WGD was found to occur early, before subclonal diversification but after the acquisition of driver alterations that induce CIN. The timing of WGD across subtypes was estimated, based on the fraction of mutations occurring post-WGD compared to pre- and post-WGD combined. On average, TNBC samples exhibited complex clonal WGD events due to a wider variance compared to HR + /HER- and HR + /HER2 + tumors (Fig. 2c), whereas HER2 + tumors revealed relatively early WGD events within subtypes.

**Clonality and timing of driver events within subtypes**. Alterations in well-known cancer genes or SMGs were frequently clonal and occurred before WGD, which suggests their involvement in tumorigenesis. In the TNBC subtype, these alterations included mutations in *CDK6*; amplification of *MYC, CCND1*, and *KIT*; and deletion of *DUSP4, LEPROTL1, NRG1, WRN, MXI1, PIK3R1, RB1*, and *STK11* (Fig. 3a, b). Other genes that were subject to subclonal involvement, suggesting late alterations in tumor progression, included *ATM* and 22q arm(deletion). In the HER2 + subtype, the mutation in *APC* and amplification of *CCNE1, MDM2, ERBB2, AKT1, EGFR, H3F3B*, and *CDK12* were early clonal alterations (Supplementary Fig. 2a, b); while deletion of *CDKN2A* and *CDKN2B*, along with 9p deletion, were late events. In the HR + /HER2 + subtype, the mutation in *TGFBR2, ACVR1B*, and *KIT*, as well as amplification of *EGFR, FGFR1, ERBB2, EIF4EBP1*, and *CDK12*, occurred as clonal alterations before WGD (Supplementary Fig. 3a, b). Notably, deletion of *DUSP4, LEPROTL1, NRG1*, and *WRN* within this subtype were primarily subclonal and occurred late. In the HR + /HER- subtype, the early mutation in *ATM, AKT1/2, PIK3CA*, and *STK11*; amplification of *MYC, CCND1, CDK6, MDM2, AKT2/3, EIF4EBP1, FGFR1*, and *HRAS*; and deletion of *RB1*, as well as pre-WGD deletion of 17p, 18p/q, 19p, and 22q, were also clonal (Supplementary Fig. 4a, b). Post-WGD mutations were found to occur in *APC* and *NOTCH1*. Interestingly, driver mutations in *TP53* were predominantly clonal, occurring before WGD in the HR + /HER2 + subtype, but after WGD in the HER2 + subtype. Similarly, other driver events in SMGs, including mutation in *PIK3CA* in the HER2 + subtype, while predominantly subclonal, occurred after WGD, which might result in reduced treatment efficacy.

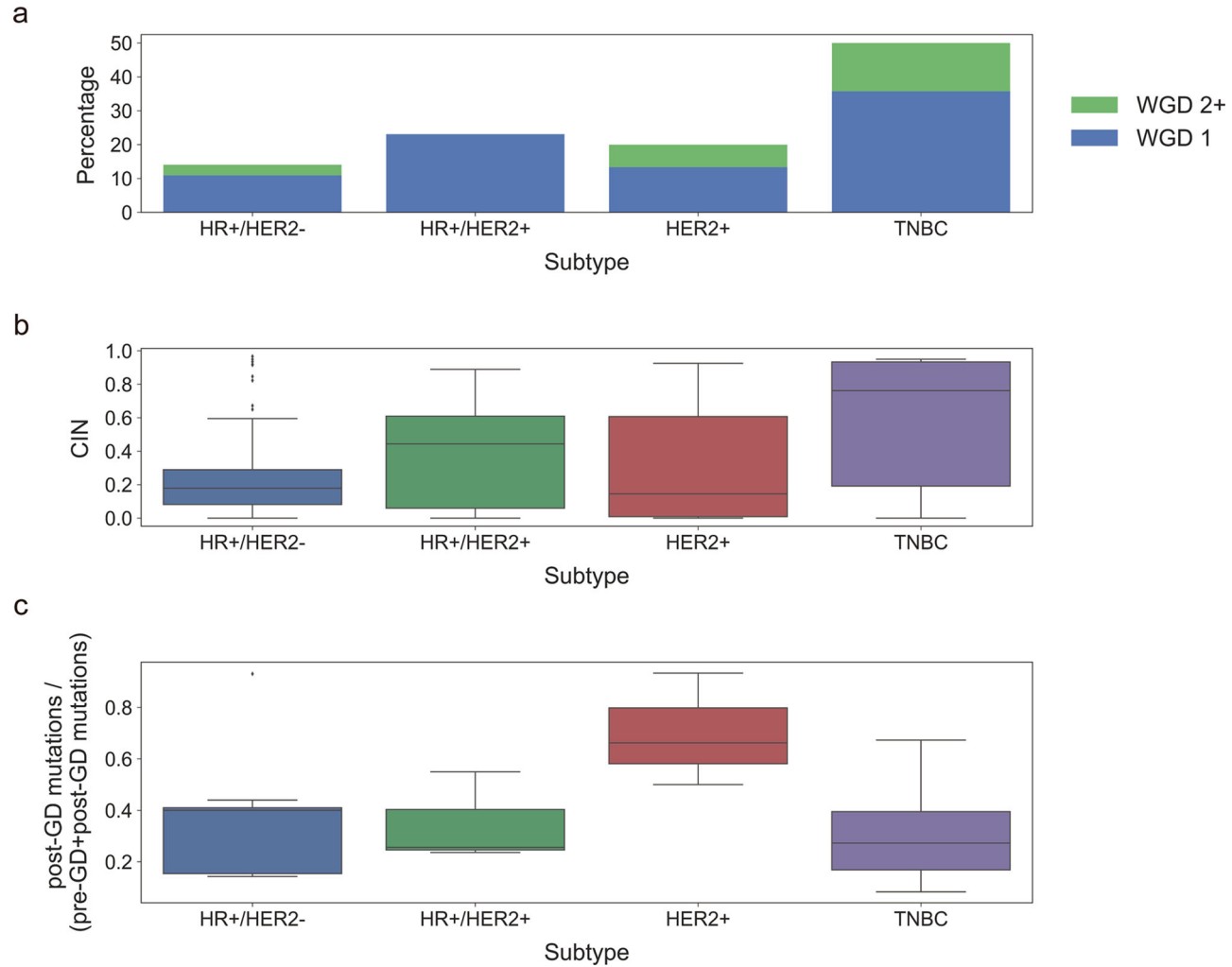

**Fig. 2 Prevalence of whole-genome doubling (WGD), level of chromosomal instability (CIN), and timing of WGD across breast cancer subtypes. a** Stacked bar plots show the fraction (y-axis) of samples harboring WGD across subtypes (x-axis). WGD 1 (blue) indicates 50% or more of the autosomal tumor genome with a major copy number of two, while WGD 2 + (green) indicates severe doubling with major copy number >2. **b** Box plots indicate the level of CIN (range, 0–1; y-axis) across subtypes (x-axis). **c** Box plots indicate the timing of WGD in each subtype. Timing was estimated based on the fraction of mutations occurring post-WGD compared to pre- and post-WGD combined. Overall, 64 HR + /HER2- (hormone receptor-positive/human epidermal growth factor receptor-negative), 13 HR + /HER2 + , 15 HER2 + , and 14 TNBC (triple-negative breast cancer) samples were used to perform analyses in **a–c**.

The mutational processes that shaped tumor progression were further explored for breast cancer subtypes. In the TNBC subtype, a large decrease in the age-related signature from the pre-WGD to the post-WGD period/stage was accompanied by an increase in the APOBEC cytidine deaminase activity (Fig. 3c). A predominance of the rate of clonal and subclonal mutations in the APOBEC signature is indicative of a long period of mutagenesis latency in the HER2 + subtype (Supplementary Fig. 2c). A similar phenomenon was observed for the APOBEC family in the HR + / HER2 + subtype, responsible for shaping the developmental trajectory of these tumors (Supplementary Fig. 3c). HR + / HER2- samples exhibited dynamic equilibrium in age and APOBEC signatures, suggesting that alternative mutational processes might dominate tumor behavior later in the tumor lifetime (Supplementary Fig. 4c).

**WGD with HRD**. To quantitatively evaluate the extent of HRD in the WGD samples, the HRD score was first calculated by combining three genomic scar scores[20]. An association test found patients with WGD to be characterized by relatively high HRD

scores ($P = 4.8 \times 10^{-8}$; one-tailed Mann–Whitney $U$ test; Fig. 4a). In each subtype, WGD-positive tumors had higher HRD scores compared to those of the WGD-negative tumors, but no difference was observed for WGD-positive tumors among subtypes (Fig. 4b). Given that 13.2% of BCTW patients exhibited HRD (score ≥ 42), the association of prevalence of the HRD phenotype with WGD status was further assessed in the subtypes. In the TNBC cohort, 35.7% of patients frequently harbored HRD that was simultaneously accompanied with WGD in 28.6% of cases (Fig. 4c). Similarly, for the HR + /HER2- and HER2 + cohort, there were more HRD cases with WGD than without WGD, in contrast to HR + /HER2 + tumors, where no difference was observed.

**WGD with alternative DSB repair processes**. To investigate whether the alternative DSB repair processes that operate in HRD cancers are associated with resulting mutational patterns, we applied the same SBS signature extraction approach (non-negative matrix factorization) to categorize indel mutational signatures[21]. De novo indel signature 2 (de novo ID2) was highly

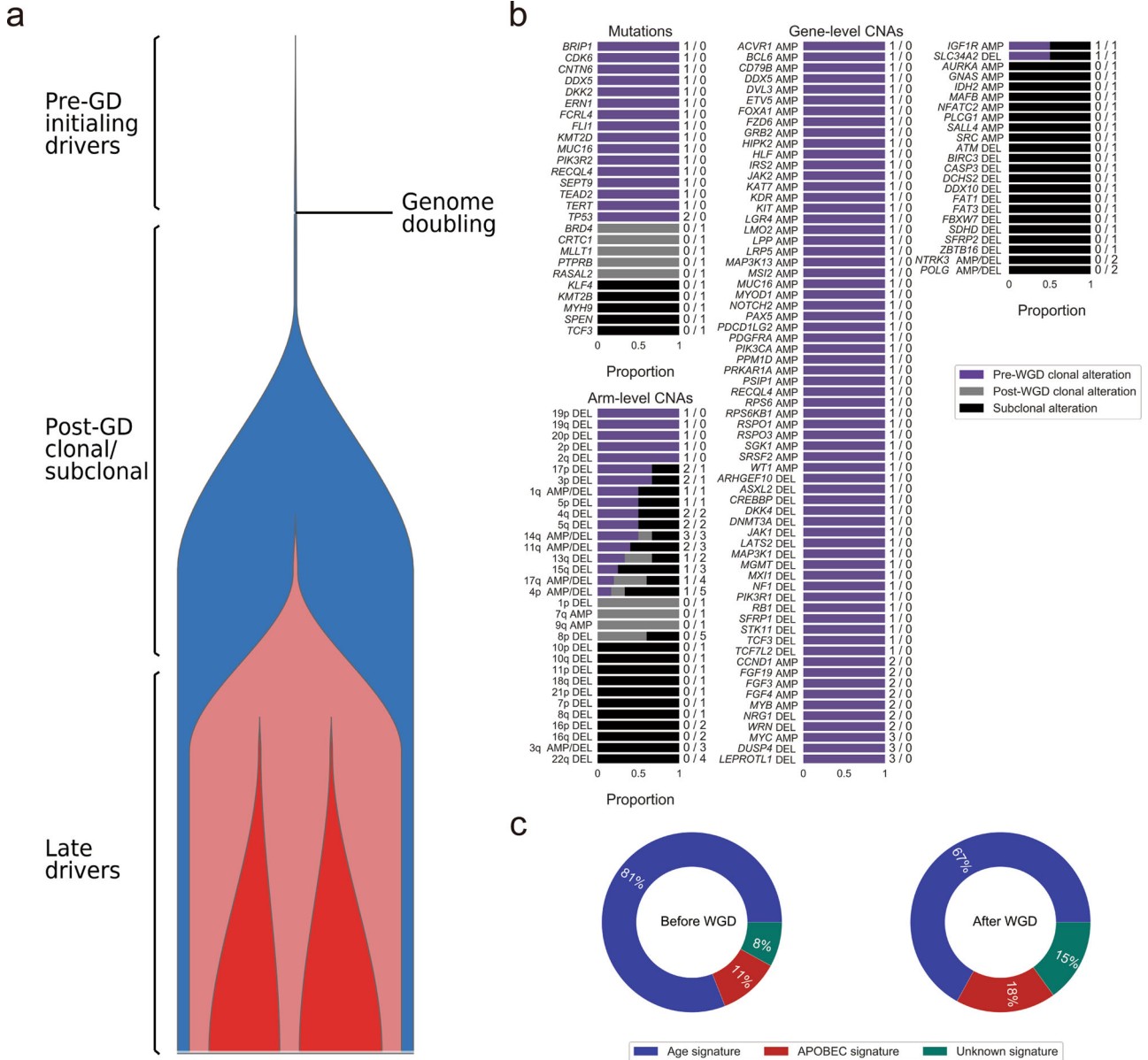

**Fig. 3 Timing of somatic events in triple-negative breast cancer (TNBC; N = 7). a** The diagram, left, of tumor evolution in the TNBC subtype shows the approximate timing of genomic alterations with respect to the cancer's lifetime. **b** The timing of mutations and copy number events is shown as bars indicating whether the events are clonal or subclonal. Clonal mutations and chromosome-arm events are further designated as early or late with respect to whole-genome doubling (WGD). The number of samples harboring mutations and copy number alterations (CNAs, pre-GD and post-GD) is indicated on the right side of the bars. **c** Pie charts show the percentage of mutations for each signature, averaged across the TNBC cohort. Only genes that were mutated in Cancer Gene Census or canonical signaling pathways in the cohort are shown. AMP: amplification; DEL: deletion.

similar to COSMIC indel mutational signatures six and eight (cosine similarity 0.85 and 0.83, respectively; Fig. 5a, b). Deletion patterns in these two COSMIC signatures have been associated with the characterization of DSB repair by two distinct forms of nonhomologous end-joining activity[21]. The COSMIC indel mutational signature six displayed longer stretches of overlapping microhomology at deletion boundaries that stem from HRD, whereas the COSMIC indel mutational signature 8 revealed relatively shorter microhomology at deletion boundaries. Interestingly, the pattern of the de novo ID2 was also similar to the mixtures of COSMIC indel mutational signatures six and eight (Fig. 5b). Also, patients with HRD were linked to a high contribution[22] of de novo ID2 ($P = 1.633 \times 10^{-4}$; one-tailed Mann–Whitney $U$ test; Fig. 5c). Furthermore, we found that the contribution of de novo ID2 in WGD-positive tumors was much

higher than in those without WGD ($P = 2.16 \times 10^{-2}$; one-tailed Mann–Whitney $U$ test; Fig. 5d). We further performed a control test to rule out the possibility that a higher contribution of de novo ID2 is caused by disproportionate numbers of indels in the WGD samples. Our results displayed no statistical significance in 978 out of 1000 times implying that the high proportion of indels in the WGD samples had no role to play in the contribution of de novo ID2 in WGD.

## Discussion

WGD potentially aggravates CIN and accelerates cancer genome evolution, which has previously been shown to be associated with poor prognosis and drug resistance[3]. In this study, it can be hypothesized that WGD might explain distinct genomic

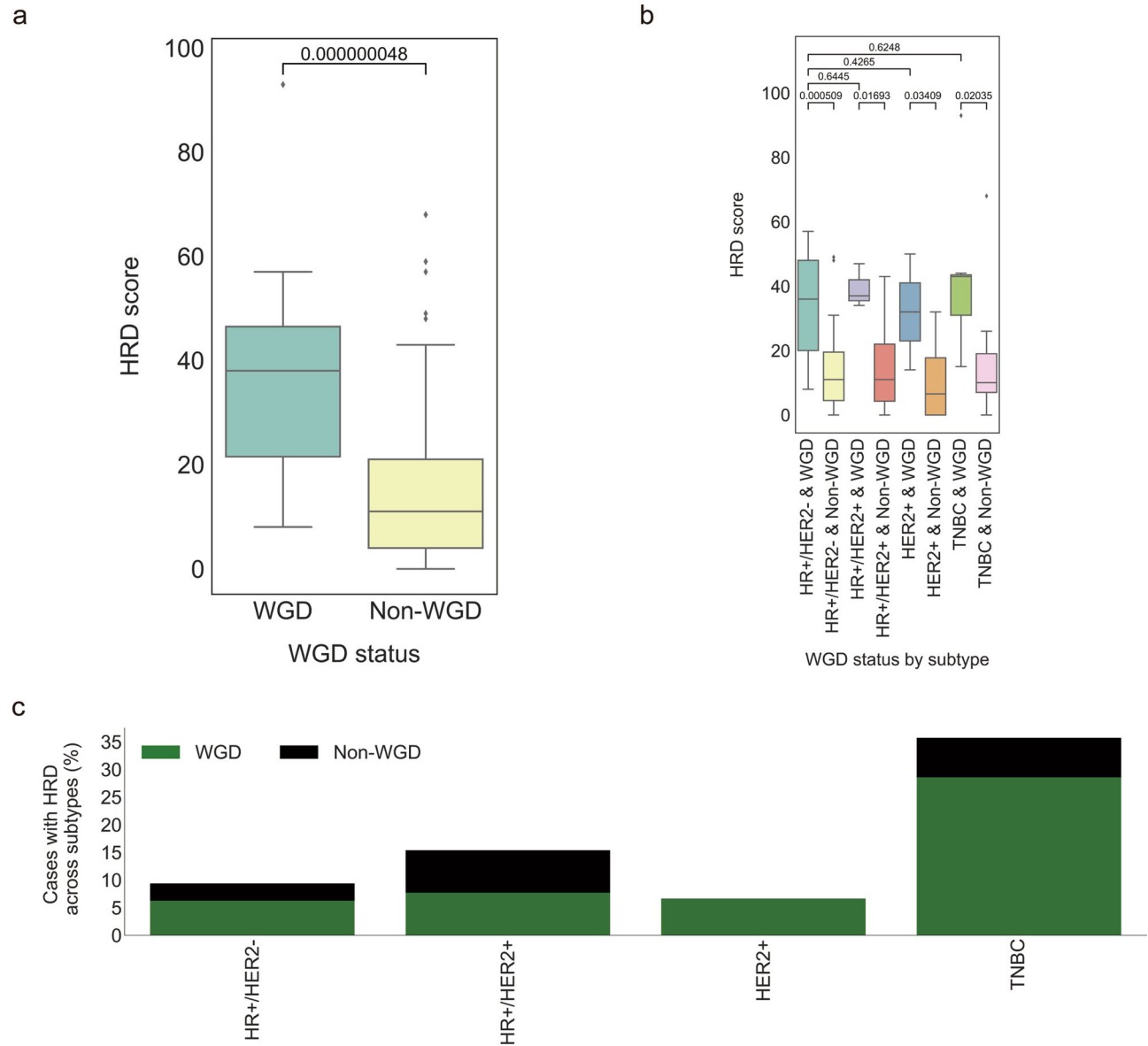

**Fig. 4 Homologous recombination deficiency (HRD) in whole-genome doubling (WGD) cancers. a** Box plots indicate the difference in HRD score (y-axis) across WGD and non-WGD tumors (x-axis). **b** Box plots indicate the difference in HRD score (y-axis) across subtypes by WGD status (x-axis). One-tailed Mann–Whitney U tests were performed to calculate the p-values for the comparison of medians in **a** and **b**. **c** Percentage of cases with HRD phenotype, defined as high HRD score (≥42) in tumors, across breast cancer subtypes (x-axis) stratified by WGD status. Overall, 64 HR + /HER2- (hormone receptor-positive/human epidermal growth factor receptor-negative), 13 HR + /HER2 + , 15 HER2 + , and 14 TNBC (triple-negative breast cancer) samples were used to perform analyses in **a–c**.

complexity across breast cancer subtypes. The prevalence of WGD displayed remarkable variability among subtypes, with its effects mostly observed in TNBC tumors. Moreover, eight somatic alterations frequently altered in the WGD samples were found across breast cancer subtypes. On average, TNBC exhibited more complex clonal WGD events when compared to other subtypes, as did HER2 + tumors with relatively early WGD events. Furthermore, TNBC samples had high levels of CIN, and CIN levels varied widely in HER2 + tumors. These findings could be the initial indication that WGD primarily affects TNBC and HER2 + tumor progression, which might change the practice of clinical care, and provide insight into breast cancer biology.

Investigation of early somatic events might assist in assessing their evolutionary impact on breast cancer in terms of both the WGD and mutational processes. The timing of cancer driver events determines their involvement in tumor initiation or progression. The clonality of drivers may facilitate therapeutic decision making, as the accumulation of subclonal alterations in a proportion of cells may result in reduced drug efficacy. However, the underlying history of deletions is complex and uncertain, hence not identifiable except with additional strong assumptions about the CN paths that are allowable. Therefore, we assume that the current copy number paths from sequencing data can be used to uncover and determine the timing of mutation and copy number alterations, based on a prior study[23]. *FGFR1* amplification, an early clonal somatic event in hormone receptor-positive tumors, conferred antiestrogen resistance to ER + breast cancer[24]. Treatment regimens by prioritizing on drug targets with combinations of ER and FGFR antagonists might improve the robust and uniform treatment response by inhibiting their binding to DNA in early ER + tumor progression. Similarly, the amplification of *EIF4EBP1*, a downstream effector of mTOR, has been

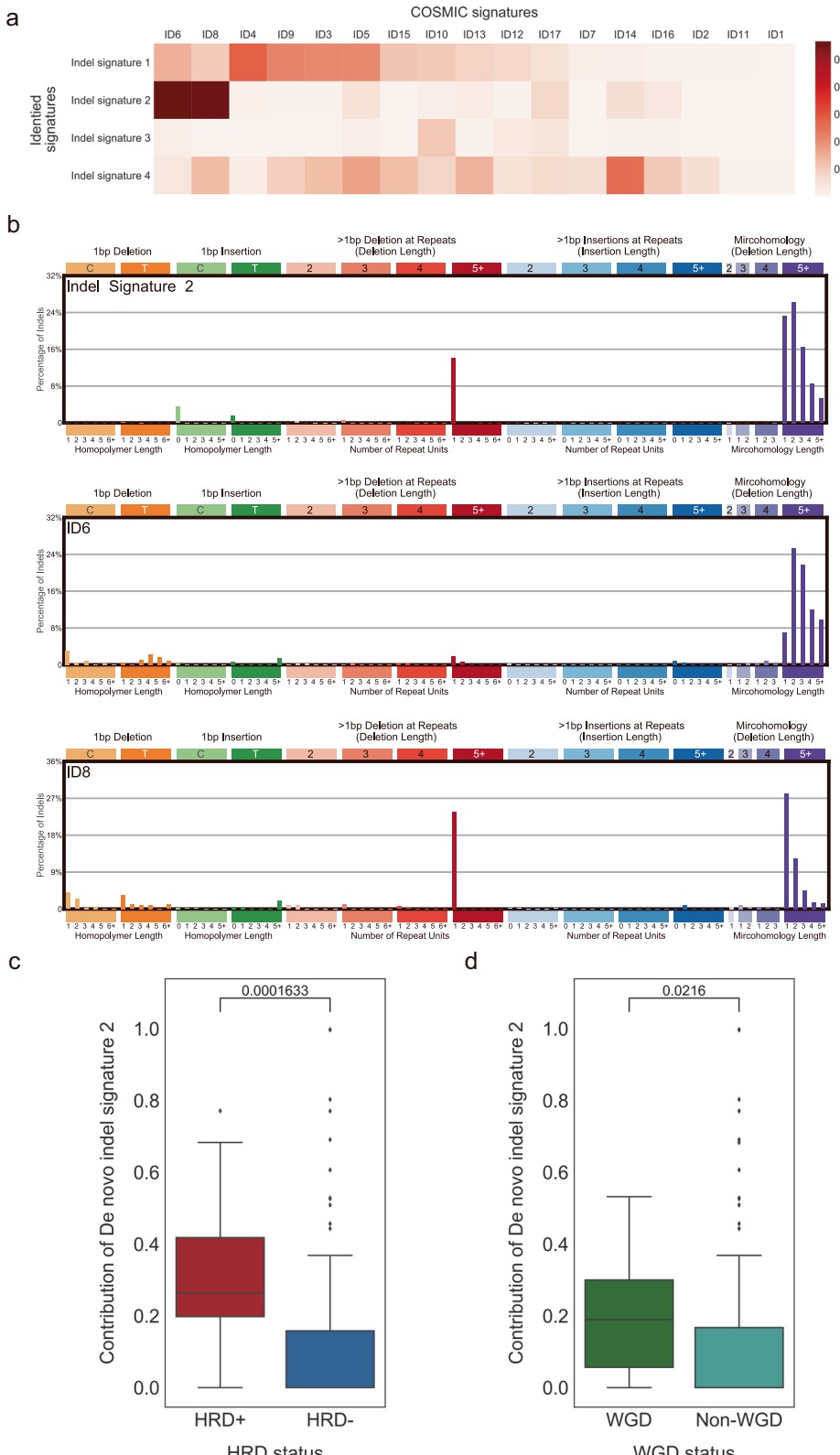

**Fig. 5 Indel mutational signatures (IDs) with homologous recombination deficiency (HRD) and whole-genome doubling (WGD). a** Heatmap of the cosine similarity results for the de novo IDs (*y*-axis), coded by color. In the scale bar, the cosine similarity (range 0–1) represents the extent of similarity to a particular signature of COSMIC (*x*-axis). Among the 17 COSMIC IDs, the signatures associated with the double-strand repair by nonhomologous end-joining were the most similar mutational signatures detected in the Taiwanese population (dark red), while other signatures are of unknown cause in COSMIC or dissimilar. **b** The mutational spectrum of de novo signature ID2 and COSMIC signatures ID6 and ID8. **c** Box plots indicate the contribution of de novo signature ID2 (*y*-axis) across HRD and non-HRD tumors (*x*-axis). **d** Box plots indicate the contribution of de novo signature ID2 (*y*-axis) across WGD and non-WGD tumors (*x*-axis). One-tailed Mann–Whitney *U* tests were performed to calculate the p-values for the comparison of medians in **c** and **d**. Overall, 116 breast cancer samples were used to perform analyses in **a**–**d**.

reported to be associated with a poor response to endocrine treatment in the hormone receptor-positive cohort[25]. A combination of ER and mTOR signaling targeted therapies might suppress hormone receptor signaling for breast tumorigenesis. In TNBC, deletions of *DUSP4, LEPROTL1, NRG1*, and *WRN* occurred before WGD, in contrast to HR + /HER2 + tumors, in which they were post-WGD. It is suggested that WGD might lead to unique therapeutic decisions and serve as a precursor of late events. These alterations could inform the design of adjuvant trials in patients with WGD and further clinical and functional investigation. Furthermore, most SMGs were altered before WGD and were predominantly clonal across subtypes but were observed post-WGD in HER2 + tumors. Notably, *CDK12* exhibited similar gene amplification-driven proteogenomic patterns to *ERBB2*;[26] both were enriched in HER2 + and HR + /HER2 + tumors as clonal alterations before WGD, which should potentially be therapeutically exploited. In the mutational processes of TNBC, a large decrease in the proportion of samples with a common age-related signature was accompanied by an increase in APOBEC-mediated mutagenesis during tumor development. Targeting the activity of APOBEC might help limit subclone diversification. These findings may have important implications for understanding the specific molecular pathogenesis and therapeutic control in breast cancer.

Although our cohort was not enriched in bi-allelic BRCA1/2 alterations, the HRD scoring measurement has the ability to identify additional patients, beyond BRCA deficiency, with high sensitivity to platinum agents or PARP inhibitors[27]. We quantitatively evaluated the extent of HRD in the WGD samples. Our observation showed that WGD-positive tumors were associated with HRD for all subtypes, and thus all might benefit from HRD-related treatment. In TNBC subtypes, patients frequently harboring WGD displayed the mutational characterization of the HRD phenotype. Therefore, future clinical trials could incorporate the WGD status for baseline treatment stratification, as several studies have suggested that some (wild-type BRCA) tumors also respond to platinum agents or PARP inhibitors, even in sporadic TNBCs[28–31]. It is hoped that such effort will help to diminish therapeutic vulnerabilities in breast cancer outcomes in the future.

This study provides accumulating evidence that the presence of a mutational signature of HRD can also be employed for optimal treatment decisions of cancer patients[32]. As COSMIC substitution signature three does not provide a precise assessment of HRD[33,34], we focused on the contribution of deletion patterns attributed to HRD, which exhibit the characteristics of non-homologous end-joining[6,7]. Given that de novo signature ID2 was identified as a mixture of COSMIC indel mutational signatures six and eight, it suggests that two distinct forms of non-homologous end-joining activity are operative in patients with HRD[21]. Furthermore, we observed that higher contribution of de novo signature ID2 is associated with WGD in BCTWs. However, the higher contribution of de novo ID2 could originate from the disproportionate numbers of indels in the WGD samples. To rule out this possibility, a control test between the WGD and pseudo non-WGD samples was employed and supported our finding. It was further observed that signature ID2 is a predictor of the interplay between HRD and WGD. Taken together, this finding might provide avenues of assessment of both WGD and HRD phenotypes for the rendering of therapeutic decisions for individual cancer patients.

In this study, we found that WGD might exhibit distinct genomic complexity across breast cancer subtypes, and highlights the association between HRD and WGD through an extensive analysis using WES data from BCTW patients. The impact of WGD and the timing of genome abnormalities across subtypes of breast cancer, which have not been characterized in the Asian cohort before, are explored in this study. In the BCTW cohort, we observed the subtype heterogeneity of WGD in Asian populations. This can be further validated in other populations, in the future. Our work has outlined the subtype specificity of WGD and CIN in BCTW patients and provided insights into the genomic basis of de novo ID2 linked to WGD and HRD in breast cancer. Future studies with suitable cell lines, which contain both the diploid and polyploid subclones with HRD phenotype from human breast cancers in four subtypes, would be needed to support our hypothesis. These suitable cell lines would be isolated at different passages and sequenced to validate the findings in this study. However, future cell line studies would be out of the scope of the current study. Such molecular-level findings may shed light on the understanding of tumorigenesis in breast cancer, and help develop personalized treatment strategies in an important step toward the complete cure of the disease.

## Methods

**Study population and specimens.** Subjects recruited for this study included a subset of clinically diagnosed breast cancer patients in Taiwan who underwent surgical resection at four different hospitals in Taiwan (Lotung Poh-Ai Hospital, Cathay General Hospital, Kaohsiung Medical University Hospital, and Cheng Ching Hospital). The study protocols were reviewed and approved by the Institutional Review Boards of these hospitals, and informed consent was obtained from all patients for conducting WES and corresponding analyses. Fresh-frozen samples were collected from tumor and matched adjacent normal tissues of 104 patients with breast cancer. Additional paired normal-tumor samples that were formalin-fixed and paraffin-embedded were available from a set of 12 breast cancer patients. Breast cancer subtypes were identified by immunohistochemistry and fluorescence in situ hybridization. To obtain BCTW-specific genetic features, the WES data was compared with sequences obtained from the TCGA database. TCGA cohorts without mutation or race information were excluded from the downstream comparisons via postfiltering strategies based on the TCGA MC3 project, leading to a total of 723 cases including Caucasians (526 cases, 72.8%), African Americans (141 cases, 19.5%), and Asian Americans (56 cases, 7.7%); linked clinical data for the TCGA cohorts was recorded in the cBioPortal for Cancer Genomics[35,36].

**Exome capture, library construction, and sequencing.** For the generation of standard exome capture libraries, we used the Agilent SureSelect XT Reagent Kit protocol for an Illumina Hiseq paired-end sequencing library (catalog#G9611A). In all cases, the SureSelect XT Human All Exon Version 6 (60 Mb) probe set was used. We used 1000 ng genomic DNA to construct each library. Each adapter-ligated sample was purified using Agencourt AMPure XP beads (Beckman Coulter, Brea, CA, USA) and analyzed on a Bioanalyzer DNA1000 chip. A total of 750 ng of the sample was prepared for hybridization with the capture baits, and the sample was hybridized for 90 min at 65 °C, captured with Dynabeads MyOne Streptavidin T1 beads (Life Technologies, USA), and purified using Agencourt AMPure XP beads. We used the Agilent protocol to add index tags by post-hybridization amplification. Finally, all samples were sequenced on an Illumina Hiseq4000 instrument using the 150PE protocol.

**Sequencing data processing.** To compare with TCGA cohorts, the analytic pipelines were run with similar parameter values as that of TCGA, based on the GDC data user guide organized by its institutional research network. The processed read pairs were mapped to the human reference genome (hg19) using BWA-MEM (v.0.7.15)[37]. The Picard module (v.2.6.0) was utilized to sort BAM files containing the sequence alignment data in binary format. Duplicate reads were marked for exclusion in a subsequent analysis using the MarkDuplicates tool in Picard. The resulting BAM files were processed with the Genome Analysis Toolkit Best Practices workflow (v.3.7) to correct mapping and sequencing errors[38]. First, indel realignment was performed around the Mills and 1000 Genomes gold standard INDELs to improve the alignment accuracy. Then base quality score recalibration was conducted to assign an accurate confidence score to each base using known variants in dbSNP138 and the Mills and 1000 Genomes gold standard INDELs[39]. To improve downstream variant detection, we added 100 bp-interval padding to confirm all reads within and outside the targeted region.

**Variant calling.** Following the TCGA DNA-Seq analysis pipeline, somatic SNV and indel calling was conducted using MuTect2 (v.3.7.0) from both the tumor and matched normal whole-exome samples in targeted exons[40], utilizing the COSMIC and dbSNP138 as reference sites of known somatic and germline mutations[18]. To confirm all calls within the targeted region, the initial SNV and indel calls were extracted using SelectVariants. The standard filter settings were the same as the TCGA MC3 project[36]. To reduce the false-positive calls, we constructed a Panel-of-Normals as filters of

contamination and miscalled germline mutations using MuTect2 on all the normal sample genomes. We removed those mutations from our Panel-of-Normals if they were present in more than one normal sample. Furthermore, the likely 8-oxoguanine error variants caused by excessive oxidation during sequence library preparation were removed by the D-ToxoG tool[41]. Artificial bias on the PCR template strand and bias on the forward/reverse strand were removed by DKFZ Bias Filter. SNVs and indels were annotated, and a Mutation Annotation Format file was produced using Variant Effect Predictor (v.89)[42] and a vcf2maf.pl script. Somatic calls that met the following criteria were used for downstream analyses: (1) variants with population allele counts ≤16 across at least one Exome Aggregation Consortium non-TCGA subpopulation (v.0.3.1);[43] and (2) variants found in the Taiwan Biobank database with population allele frequency ≤0.0001[44].

**Discovery strategy for SMGs.** To discover SMGs, samples with a high number of aberrations (>400 SNVs and indels) were excluded to reduce the artefactual sensitivity to high background mutation rates. A combination of four approaches was implemented to create a consensus list of SMGs: (1) dNdScv (v.0.0.0.9), a maximum-likelihood dN/dS method, quantifies selection of missense, nonsense, and splice mutations, at the level of individual genes, groups of genes, or at the whole-genome level in cancer and somatic evolution[14]. dNdScv was run using default parameters. Genes with a $q$ value < 0.1 were considered as SMGs. (2) OncodriveCLUST (v.1.0.0), a method used to identify oncogenes with a bias towards mutation clustering for genes with nonsynonymous mutation more than expected of synonymous mutations within the protein sequence[17]. OncodriveCLUST was run using default parameters. Genes with a $q$ value <0.05 were considered as SMGs. (3) OncodriveFML (v.2.0.3), a method to analyze the pattern of somatic mutations across tumors in both the coding and noncoding genomic regions to identify signals of positive selection, and therefore, their involvement in tumorigenesis[16]. Genes with a $q$ value <0.25 were considered as SMGs. (4) 20/20+ (v.1.1.3), a machine learning-based and ratiometric method that classifies genes as oncogenes and tumor suppressor genes from somatic mutations[15,45]. Genes with a $q$ value <0.05 were considered as SMGs. All the significance thresholds are based on a previous study[46]. In addition, we required these genes to be expressed in human breast tissue according to The Human Protein Atlas[11].

**Mutational signature analysis.** The final portrait of mutations was determined by the duration of exposure to each mutational process in patients. Mutational signatures were deciphered from the substitution context defined by 96 mutation types, composed of flanking bases as triplet motifs. The contributions of each mutational signature were quantified to deduce its association with mutagenic processes, such as ultraviolet light exposure, carcinogens, and aging[22]. To characterize the SBS mutational signatures originating from the accumulation of historic mutagenic activity in BCTWs, we utilized non-negative matrix factorization to extract de novo SBS mutational signatures present in these tumors by utilizing the R package NMF (v.0.21.0)[47]. Motif matrices were extracted using MutationalPatterns (v.1.2.1)[48]. The inferred mutational signatures were then compared to the curated catalog of 30 SBS mutational signatures referenced in COSMIC using cosine similarity[18]. Most signatures with cosine similarity >0.85 corresponded to one or a mixture of known signatures.

All indels were classified as deletions or insertions, while single base indels were subdivided according to the homopolymer length in repeat sequences of sites of occurrence. Longer indels were further classified as occurring at repeats or with overlapping microhomology at deletion boundaries, based on their size[21]. Taken together, the primary classification of indels comprised 83 classes. The extraction of indel signatures and assessment of each signature to each cancer genome was similar to that used with SNVs and visualized by using SigProfilerMatrixGenerator[49]. To rule out the possibility of the higher contribution of de novo ID2 caused by disproportionate numbers of indels that are called from the WGD samples (Supplementary Data 1), a control test was performed. From each WGD sample, we randomly sampled indels consistent with the median indel count of the non-WGD cohort, treating it as a pseudo non-WGD sample, and estimated the contribution of de novo ID2. We subsequently checked if these control samples showed a lower contribution to de novo ID2 than the WGD samples using the one-tailed Mann–Whitney $U$ test. The above computations were further repeated 1000 times to compute the probability.

**Allele-specific copy number profiling.** Allele-specific copy numbers were called using the FACETS algorithm (v.0.5.2) for WES data[50]. Reference and variant allele read counts were extracted from the tumor and matched normal BAM files at germline polymorphic sites, which are catalogued in the dbSNP and 1000 Genomes databases with base quality >20 and mapping quality >15, and only sufficiently covered regions with >25 read counts were considered. Heterozygous single-nucleotide polymorphisms in normal samples were used, and allele-specific copy number profiles for matched tumor samples were analyzed with default settings. Allele-specific copy number profiles were adjusted by estimated tumor purity and ploidy to improve the accuracy. To establish the associations between CNAs and subtype in breast cancer, we analyzed both the arm-level and focal copy number calls based on GISTIC2 (v.2.0.23)[51]. The copy number level per gene was also defined by default settings. The gene-level amplification/deletion values produced by GISTIC2 were also used in this analysis, considering only high-level amplification (+2) and deep deletion (−2).

**Assessment of BCTW sample quality.** To improve the somatic mutation detection power for downstream analyses, we performed two additional procedures. The DeTiN algorithm was applied to rescue the false-negative calls due to potential tumor-in-normal contamination[52]. FACETS and ABSOLUTE were utilized to assess the tumor purity and determined that most tumor samples had purity estimates between 30 and 57%, with a median of 39%;[53] while the median sequencing depth of WES was more than 200 in both tumor and matched normal tissues with at least 20x depth for 90% of target exons. Prior studies have indicated that a purity of 24% is enough to detect clonal events[54]. These steps aimed to confirm that the mutational differences originated from genomic racial distinctions and the timing and clonality of somatic mutations were associated with WGD rather than the confounder of tumor-in-normal contamination and low purity.

**WGD and CIN.** We identified the presence of WGD in the tumors of characterized breast cancer patients using an analysis of allele-specific copy number, which counts maternal and paternal alleles based on the sequencing coverage and genotypes of germline single-nucleotide polymorphisms. In heterozygous regions of a diploid cancer genome, there is one copy of each maternal and paternal allele. However, in a genome-doubled tumor, the number of copies of the more frequent allele (major copy number (MCN)) should be elevated across a substantial fraction of the cancer genome. Patients were considered to have undergone WGD if more than 50% of their autosomal genome had an MCN derived from FACETS greater than or equal to two[4]. Two-tier classification used the terms WGD 1 for MCN equal to two and WGD 2+ for MCN greater than two. CIN is a broad concept that encompasses a wide range of chromosome-level abnormalities. CIN is defined as the percentage of the genome in length affected by CNAs[55] and is given by:

$$CIN = \frac{\sum_{i=1}^{n} u_i}{L}$$

where L is the total length of the autosome and $u_i$ represents the altered length in CAN $i$.

**Clonality of somatic alterations and timing of WGD.** To infer the clonality, the estimates of purity and allele-specific copy number from FACETS were used as input for the Absolute module to obtain the cancer cell fraction for all somatic mutations with corresponding variant allele fractions in all tumor samples. Somatic alterations were further timed relative to WGD using methodology adapted from the previous work[23,56,57]. Clonal mutations in regions of overlapping copy number events were timed as early or late with respect to WGD. Mutations were classified as early if the mutation copy number was >1 and the MCN ≥ 2. Late mutations were called with the mutation copy number ≤1 and MCN ≥ 2. Clonal mutations that could not be timed were classified as untimed, while any subclonal mutations were considered as late events after WGD. The timing of WGD was estimated based on the fraction of mutations occurring post-WGD compared to pre- and post-WGD combined[23].

The FACETS algorithm was also utilized to estimate the cancer cell fraction to distinguish the clonal and subclonal CNAs. Clonal arm-level amplification was timed as early and late with respect to WGD by an average ratio of mutation copy numbers according to the overlapping mutations on a given arm. Arm-level amplification was classified as early if the majority of mutation copy numbers was ≤1. Late arm-level amplification was called with the majority of mutation copy numbers >1. Clonal arm-level amplification was classified as untimed if there were insufficient mutations (<3) mapping to a given arm, while any subclonal arm-level amplification was considered as a late event after WGD. For clonal arm-level deletion, it was timed relative to the loss of heterozygosity (LOH) status in WGD tumors. Arm-level deletion showing LOH was classified as early. Clonal arm-level deletion without LOH was called as late, while any subclonal arm-level deletion was considered as a late event after WGD.

**HRD analysis.** A DNA-based predictor (HRD score) derived from the unweighted sum of HRD-LOH score[58], the number of telomeric allelic imbalance[59], and the large-scale state transitions score[60], was applied to measure the underlying tumor HRD[20]. The HRD-LOH score is defined as the total number of LOH events longer than 15 Mb without covering the whole chromosome. The number of telomeric allelic imbalance score is the sum of regions with the allelic imbalance that extend to the telomeric end of a chromosome without crossing the centromere. The large-scale state transitions score is the number of chromosomal breaks between the adjacent regions longer than 10 Mb, with a distance between them shorter than 3 Mb. Based on allele-specific copy numbers for each region, the HRD score was calculated using scarHRD[61]. A predefined HRD threshold of 42 (≥42) was employed to distinguish the HRD phenotype from nondeficient tumors[20].

**Statistics and reproducibility.** Mutational burden, HRD score, and the contribution of de novo signature ID2 related to WGD was analyzed using a one-tailed Mann–Whitney $U$ test to determine if WGD-positive tumors have a higher burden than WGD-negative tumors, as well as the association between the mutational

status of SMGs and mutation burden. Logistic regression was performed to identify relatively frequent cancer gene alterations in the WGD samples. All statistical analyses were performed using R version 3.5.1. Sample sizes are included in each figure legend. The allelic ratio plots and sequencing quality metric for assessing WGD analysis are shown in Supplementary Figs. 5–26 and Supplementary Data 2. The bootstrapping data for 95% confidence interval of mutational frequencies in each population (Fig. 1b) are shown in Supplementary Data 3.

**Reporting Summary**. Further information on research design is available in the Nature Research Reporting Summary linked to this article.

## Data availability

Raw whole-exome sequencing data used in this study can be found in the NCBI database under the BioProject accession PRJNA729775. The molecular and clinical data of breast cancer patients from the TCGA cohort are available in the following repositories: TCGA BRCA: https://portal.gdc.cancer.gov/ (MuTect2 Mutation Annotation Format file) and cBioPortal: https://www.cbioportal.org/ (clinical data).

## Code availability

All of the tools used in this study are publicly available. Statistical and other analyses were performed using software R (version 3.5.1) and described in the method section.

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

## Acknowledgements

We would like to thank TCGA, COSMIC, Exome Aggregation Consortium, Taiwan Biobank, and The Human Protein Atlas for making their data publicly available. Finally, we thank all the participating individuals for their contribution to this study. This work was supported by the Center of Biotechnology, National Taiwan University, Taiwan [grant number GTZ300].

## Author contributions

Y.-C.C. (Yo-Cheng Chang), M.-H.T. and E.Y.C. designed the study. Y.-H.L., M.-H.T. and L.-C.L. performed the experiments. Collection of specimens was coordinated by C.-C.H., H.-T.Y., M.-F.H., Y.-C.C. (Yuan-Chiang Chung), S.-H.T. and K.-J.C. Clinical data was organized by C.-C.H., H.-T.Y., Y.-H.L., T.-P.L. and C.-S.H. Data analyses were conducted by C.-H.W. and C.-S.H. C.-H.W. performed statistical analyses. C.-H.W., C.-S.H., A.C., M.-H.T. and E.Y.C. wrote the manuscript. All authors read and approved the final manuscript.

## Competing interests

The authors declare no competing interests.
