## [Transparent Peer Review File · Communications Biology]

Reviewers' comments:

Reviewer #1 (Remarks to the Author):

By analyzing the whole-exome sequencing of 115 breast-cancer patient samples from Taiwan. Wu et al. here describe the heterogeneity and homogeneity in the timing and dependencies of somatic aberrations across the disease subtypes. In particular, the authors focus on somatic alterations and related features that correlate with whole genome doubling, claiming that homologous recombination deficiency plays a critical mechanistic role. This study presents some interesting observations and provides a cohort that may facilitate further studies when combined with other datasets. However, I am not convinced that the current form of the study delivers solid mechanistic connections in BC genomics. Below are my concerns about the methodology, overall statistical assessments, as well as the interpretation of these results.

General Concerns:

Timing of Genomic Changes relative to WGD:

The study uses threshold-based criteria (somehow arbitrary) to determine the relative ordering of mutation and copy number variants to the whole genome doubling events. However, the sequencing data is associated with inherent variability in allelic depth, and the error rate or reliability of such an inference is not presented. Besides, existing literature in this area has emphasized that for deletions, the underlying history is not identifiable except with additional strong assumptions about the CN paths that are allowable. At least, it needs to be stated explicitly in the paper about these assumptions, and discussions of the uncertainty and complexity of these relative ordering are necessary. Besides, the sample size is a concern; for the WGD group, there are 19% of patients (~ 22) with WGD. I would suggest that authors focus on generating hypotheses using these data rather than making strong conclusions.

The Association between WGD and HRD:

It is expected that tumors with high overall ploidy tend to show catastrophic genomic copy number patterns, as WGD allows the cell to sample different combinations of CN landscape that equip them with a better adaptive advantage in the microenvironment. As HRD score is calculated based on the extent of CN aberrations (that correlates with homologous recombination deficiency), it is also expected to be high under WGD. Therefore, I am not convinced that such a quantitative association causally relates the WGS to homologous recombination deficiency in this disease.

Authors try to find the enrichment of LOF of HR-related genes to support such a conclusion.

However, they treat variants of unknown significance as a potential loss of function. Furthermore, the statistical significance is not reported of observing ~5 genes having early LOF in ~5 patients with WGD and HRD using the above definition of deleterious mutations.

Indel Signature:

The strongest evidence probably comes from the indel signature (cosmic ID6 and 8 in particular) that supports potential homologous recombination defects in cases with WGD. Although WGD cases seem to contribute more to these indel signatures, it also could be due to the disproportionate numbers of indels called from WGD cases than from those without WGD (as authors also reported that WGD cases have higher mutation burden). Authors need to present the number of indels called in these samples. To rule out the possibility that I just mentioned, they can perform a "control" test by randomly sample indels from the WGD cases (to the same number as in the non-WGD cases), treating them as "pseudo" non-WGD cases, perform the signature analysis and then check if these control samples show a significantly lower contribution to the signature identified.

Recurrent alterations associated with WGD:

The genes reported to be associated with WGD, i.e., EIF4EBP1, FGFR1 (with recurrent amplification reported in the paper) and DUSP4, LEPROTL1, WRN, NRG1 (with recurrent deletion), are all located on

chr8p. In fact, they are quite close to each other on this arm. It is a bit surprising to see that in a clustered genomic region, both recurrent amplifications and deletions are observed. As with whole-exome sequencing (WXS), we are less powered to call focal copy number changes than with WGS, I suggest that authors provide the visualization of GISTIC results in the extended data. This also raises the concern that under WGD, the detection of recurrent changes could be heavily influenced by systematic biases. As this paper is not focusing on devising the computational methodology, I would recommend turning down the claims associated with these discoveries.

The only other such gene, MYC, is located on chr8q. This gene seems to be upregulated in various cancer types with genome doubling. Do authors want to point out any specificity of this pattern in their cohort? If not, I would consider this as a known general pattern.

Missing sequencing quality metrics of each sample:

Authors show that most SMG occur before WGD in most subtypes, but not in HER2+. How much of these are attributable to differences in sequencing depth or sample purity? In fact, the sequencing and sample quality metrics are not presented in this paper.

Plots of Copy Number Profiles:

Authors need to add plots showing the read depth ratio (between tumor and normal), as well as the allelic ratio (BAF), in the supplementary information, for readers to assess the validity of the analysis and quality of the sequenced samples.

Specific comments to improve analysis and clarity:

Figure 1a, on the right, I would suggest using box or bar plots with 95% interval reported for each population (with bootstrapping using the same sample size as the studied cohort). This is to statistically confirm the differences in mutational frequencies between the BCTW cohort and others.

Figure 1b and 5a, it would be clearer by marking in the figure that the x-axis is referring to cosmic signatures and a scale bar for the similarity. The heatmap needs to be sorted somehow as well.

The discrepancy between Figure 2c and line 184: mutations are less prevalent post GD for HER2+ subtype, this seems to suggest a "late" rather than "early" (stated at line 184) acquisition of GD.

Figure 3 and Extended Data Fig. 2-4. Colouring is misleading and needs to be adjusted. For example, blue was used for both pre-GD and post-GD in figure 3a but was only referring to pre-GD in figure 3b. I would also suggest to revise the legend of figure 3b, to indicate that the right side of the bars in figure 3b shows the number of patient samples (instead of saying "frequency") falling in each timing group (pre-GD and post-GD). For gene-level and arm level CNA in figure 3b, authors should also note if they are amplification and deletions.

Reviewer #2 (Remarks to the Author):

Summary: The present manuscript evaluates the genetic architecture of breast cancer arising from Taiwanese patients. Whole exome sequencing of paired tumor normal specimens was performed which illustrated mutational events that are similar to those which have been reported in TCGA and other studies. Subsequent analyses of whole-genome doubling, shows that this event is most common in terminal negative breast cancer (TNBC) and is associated with chromosome instability. Using allelic balance the timing of different variants is mapped relative to the whole-genome doubling. The presence of whole-genome doubling is associated with homologous-recombination repair deficiency which is inferred by genomic scarring. Lastly, the whole-genome doubling is found to correlate with a specific indel score.

Critique: While the study has merit, there is very little in the work which is new and much of the manuscript is confirmatory. Potential unique elements are not clearly articulated, and significance specific to Taiwanese patients and the manifestation of breast cancer is not clear. A few points are provided below:

1. The number of cases is relatively limited for sub-type specific interrogation. There are only 14 TNBC patients in the cohort, so sub-stratifying further limits any statistical power particularly when correcting for other variables (e.g. grade)
2. Whole genome duplication and increased chromosome instability is already known to be over-represented in TNBC relative to other subtypes of breast cancer. Similarly the mutational signatures are largely consistent with what is known for sporadic breast cancer.
3. The clonal assessment of the mutation events is interesting, but given the small numbers of events it would seem to be impossible to make a rigorous conclusion from pre vs. post whole genome doubling. In general the presentation would benefit by clearly disclosing the number of cases being analyzed and how they are being pooled/interrogated. Presenting individual cases may better illustrate the point being indicated.
4. The analyses of homologous repair deficiency and INDEL signatures is similarly interesting, but increased homologous repair deficiency is already known to occur in TNBC and the powering relative to the conclusion (ie. number of tumors) needs to be disclosed. Boot strapping with other data or some other approach may be necessary to provide additional support for these conclusions. This is particularly relevant with regard to the analysis of WRN.
5. Some form of functional studies to support the associations in the study would be important to solidify the observations with reference to homologous repair deficiency and whole genome doubling as others have performed using different modeling systems.

Reviewer #1

By analyzing the whole-exome sequencing of 115 breast-cancer patient samples from Taiwan. Wu et al. here describe the heterogeneity and homogeneity in the timing and dependencies of somatic aberrations across the disease subtypes. In particular, the authors focus on somatic alterations and related features that correlate with whole genome doubling, claiming that homologous recombination deficiency plays a critical mechanistic role. This study presents some interesting observations and provides a cohort that may facilitate further studies when combined with other datasets. However, I am not convinced that the current form of the study delivers solid mechanistic connections in BC genomics. Below are my concerns about the methodology, overall statistical assessments, as well as the interpretation of these results.

General Concerns:

Timing of Genomic Changes relative to WGD:

The study uses threshold-based criteria (somehow arbitrary) to determine the relative ordering of mutation and copy number variants to the whole genome doubling events. However, the sequencing data is associated with inherent variability in allelic depth, and the error rate or reliability of such an inference is not presented. Besides, existing literature in this area has emphasized that for deletions, the underlying history is not identifiable except with additional strong assumptions about the CN paths that are allowable. At least, it needs to be stated explicitly in the paper about these assumptions, and discussions of the uncertainty and complexity of these relative ordering are necessary. Besides, the sample size is a concern; for the WGD group, there are 19% of patients (~ 22) with WGD. I would suggest that authors focus on generating hypotheses using these data rather than making strong conclusions.

Response: We thank the reviewer very much for your thoughtful comments. We mainly followed methods from Mariam et al., 2017 to determine the timing of mutation and copy number alterations. According to the reviewer's suggestion, we have also added the allelic ratio (BAF) plot and quality metric for readers to assess the error rate or reliability in Supplementary Figures 5-26 and Supplementary Data 2 (**Change #1 and #2 / line 515-517**). We agree that assumption for history of deletions is important

for timing. In keeping with this recommendation, we have added descriptions in the Discussion to highlight the assumption (**Change #3 / line 262-266**). As pointed out, generating hypothesis for the observations of WGD is more proper than making strong conclusion. We have added sentences in the Introduction and Discussion sections (**Change #4 and #5 / line 76-78 and 248-250**). The previous biological finding of BC subtypes caused by different profiles of SNVs and CNAs allows us to hypothesize that large-scale somatic events might also explain the subtype heterogeneity.

The Association between WGD and HRD:

It is expected that tumors with high overall ploidy tend to show catastrophic genomic copy number patterns, as WGD allows the cell to sample different combinations of CN landscape that equip them with a better adaptive advantage in the microenvironment. As HRD score is calculated based on the extent of CN aberrations (that correlates with homologous recombination deficiency), it is also expected to be high under WGD. Therefore, I am not convinced that such a quantitative association causally relates the WGS to homologous recombination deficiency in this disease.

Authors try to find the enrichment of LOF of HR-related genes to support such a conclusion.

However, they treat variants of unknown significance as a potential loss of function. Furthermore, the statistical significance is not reported of observing ~5 genes having early LOF in ~5 patients with WGD and HRD using the above definition of deleterious mutations.

Response: Thank you for your valuable comment. As pointed out, WGD causes the gain of genomic copy number, while the HRD score is also based on the genomic scars of loss or LOH in copy number profiles. HRD could cause chromosomal instability to induce other somatic events (Knijnenburg et al., 2018). As both WGD and HRD increase chromosomal instability, we believe that it is worthy of verifying the true state of the extent of HRD in the WGD samples to bring out the Results section 6. Therefore, we tried to quantitatively examine the profile and report relevant results to determine the extent of HRD in the WGD and non-WGD samples in BCTWs. We have added sentences to highlight this aim in the Introduction (**Change #6 / line 92-94**). As pointed out, the statistical concerns regarding the enrichment of LOF of HR-related genes to support the associations between WGD and HRD is a challenging issue. We are unable

to add more sequencing data of triple-negative patients from our cohort. Therefore, we have modified and deleted those sentences in the manuscript and focus on the evidence from indel signature based on your thoughtful suggestion (**Change #7-15 / line 58-59, 100-102, 215, 225, 288-290, 292, 314-316, 509, 721, and 730**).

Indel Signature:

The strongest evidence probably comes from the indel signature (cosmic ID6 and 8 in particular) that supports potential homologous recombination defects in cases with WGD. Although WGD cases seem to contribute more to these indel signatures, it also could be due to the disproportionate numbers of indels called from WGD cases than from those without WGD (as authors also reported that WGD cases have higher mutation burden). Authors need to present the number of indels called in these samples. To rule out the possibility that I just mentioned, they can perform a “control” test by randomly sample indels from the WGD cases (to the same number as in the non-WGD cases), treating them as “pseudo” non-WGD cases, perform the signature analysis and then check if these control samples show a significantly lower contribution to the signature identified.

Response: Thank you very much for your valuable comments. We agree with the reviewer’s comment that the number of indels called from these samples need to be shown. We have provided the number of indels for each sample in the Supplementary Data 1 (**Change #19 / line 424-431**). We appreciate this important recommendation of a “control” test that improves this study. According to your suggestion, we have assessed the contribution of the identified signature 2 in the “pseudo” non-WGD cases and found no significant difference between the original WGD and “pseudo” non-WGD cases. We have added this approach in the manuscript (**Change #16-19 / line 59-60, 240-244, 302-306, and 424-431**).

Recurrent alterations associated with WGD:

The genes reported to be associated with WGD, i.e., EIF4EBP1, FGFR1 (with recurrent amplification reported in the paper) and DUSP4, LEPROTL1, WRN, NRG1 (with recurrent deletion), are all located on chr8p. In fact, they are quite close to each other on this arm. It is a bit surprising to see that in a clustered genomic region, both recurrent amplifications and deletions are observed. As with whole-exome sequencing (WXS), we are less powered to call focal copy number

changes than with WGS, I suggest that authors provide the visualization of GISTIC results in the extended data. This also raises the concern that under WGD, the detection of recurrent changes could be heavily influenced by systematic biases. As this paper is not focusing on devising the computational methodology, I would recommend turning down the claims associated with these discoveries. The only other such gene, MYC, is located on chr8q. This gene seems to be upregulated in various cancer types with genome doubling. Do authors want to point out any specificity of this pattern in their cohort? If not, I would consider this as a known general pattern.

Response: We apologize for our poor explanation. As we mentioned in the Methods, the gene-level amplification/deletion values produced by GISTIC2 were used in this analysis, considering only high-level amplification (+2) and deep deletion (-2). We refer to the results of GISTIC2 to check the CNA status of each gene in every sample. We did not discuss focal- and am-level CNAs based on statistical tests of GISTIC2 in this study. Therefore, we provide the visualization of GISTIC2 results below. According to your suggestion, we have turned down the claims associated with these discoveries in the manuscript (**Change #20-27 / line 97, 169, 171-174, 194-196, 251-252, 266-267, 273-274, 509, 513, and 514**). Instead, we have modified those sentences to mention that these alterations of cancer genes were frequently found in the WGD samples. As pointed out, amplification of *MYC* had been reported in the WGD samples. We have toned down the description in the Abstract and Discussion sections and mentioned a similar result that we also found in our study (**Change #28 and #29 / line 58 and 277**).

Missing sequencing quality metrics of each sample:

Authors show that most SMG occur before WGD in most subtypes, but not in HER2+. How much of these are attributable to differences in sequencing depth or sample purity? In fact, the sequencing and sample quality metrics are not presented in this paper.

Response: We appreciate your valuable comments. According to your suggestion, we have provided the quality metrics for readers (**Change #2 / line 515-517**). Sequencing quality in most samples of these subtypes were similar. Indeed, purity in these subtypes existed difference (in HR+/HER2+). As a prior study reported that a purity of 24% is enough to detect clonal events (Amin-Mansour, A. et al., 2019), we have added this

information in the Methods of manuscript. Given that purity in these samples with WGD were all >24%, we believe that SMGs before WGD as clonal events could be detectable.

Plots of Copy Number Profiles:

Authors need to add plots showing the read depth ratio (between tumor and normal), as well as the allelic ratio (BAF), in the supplementary information, for readers to assess the validity of the analysis and quality of the sequenced samples.

Response: Thank you for your valuable comments. We have added plots of BAF in the Supplementary Figures 5-26 for readers to assess the analysis and quality (**Change #1 / line 515-517**).

Specific comments:

Figure 1a, on the right, I would suggest using box or bar plots with 95% interval reported for each population (with bootstrapping using the same sample size as the studied cohort). This is to statistically confirm the differences in mutational frequencies between the BCTW cohort and others.

Response: We thank the reviewer for your thoughtful comments. We have added bar plot with 95 % CIs in Fig. 1 and modified sentences in the Results (**Change #30-34 / line 144-146, 153, 682, and 687-692**).

Figure 1b and 5a, it would be clearer by marking in the figure that the x-axis is referring to cosmic signatures and a scale bar for the similarity. The heatmap needs to be sorted somehow as well.

Response: Thanks for your valuable comment. To avoid confusion, we have added the detailed information in new Fig. 1c and 5a and modified the figure legends (**Change #35 and #36 / line 682, 693, 694, 731, and 734**). According to your suggestion, we have added the scale bar of cosine similarity and labeled the x- and y-axis, referring to COSMIC signatures and identified signatures, respectively. We have also sorted the heatmap in order of COSMIC signatures highly similar to identified signatures

(similarity > 0.8). The rest order was sorted by the similarity between de novo signature 1 and the rest COSMIC signatures.

The discrepancy between Figure 2c and line 184: mutations are less prevalent post GD for HER2+ subtype, this seems to suggest a “late” rather than “early” (stated at line 184) acquisition of GD.

Response: Thank you for indicating an important point. The initial Fig. 2c was visualized to indicate the timing by "the fraction of mutations occurring pre-WGD compared to pre- and post-WGD combined". Before submission, in Fig. 2c, we thought that using "the fraction of mutations occurring post-WGD compared to pre- and post-WGD combined" might be better visualized for readers to assess the timing. However, in the previous manuscript, the fraction was not successfully changed during coding for Fig. 2c. In the revised manuscript, we have corrected this issue in new Fig. 2c (**Change #37/ line 700**).

Figure 3 and Extended Data Fig. 2-4. Colouring is misleading and needs to be adjusted. For example, blue was used for both pre-GD and post-GD in figure 3a but was only referring to pre-GD in figure 3b. I would also suggest to revise the legend of figure 3b, to indicate that the right side of the bars in figure 3b shows the number of patient samples (instead of saying “frequency”) falling in each timing group (pre-GD and post-GD). For gene-level and arm level CNA in figure 3b, authors should also note if they are amplification and deletions.

Response: We thank the reviewer for valuable comments. According to your suggestion, we have labeled amplifications and deletions, corrected the coloring and modified the figure legend in Fig. 3 and Supplementary Figures 2-4 (**Change #38 / line 711, 716, 717, and 720**).

The visualization of GISTIC results

The visualization of GISTIC results of triple-negative breast cancer (TNBC) cohort. GISTIC plots indicate the G-scores (top) and q-values (bottom) with respect to amplification (a) and deletion (b) in the chromosome regions. The green line indicates the significance threshold (q-value= 0.25).

The visualization of GISTIC results of human epidermal growth factor receptor-positive (HER2+) cohort. GISTIC plots indicate the G-scores (top) and q-values (bottom) with respect to amplification (a) and deletion (b) in the chromosome regions. The green line indicates the significance threshold (q-value= 0.25).

The visualization of GISTIC results of hormone receptor-positive/human epidermal growth factor receptor- positive (HR+/HER2+) cohort. GISTIC plots indicate the G-scores (top) and q-values (bottom) with respect to amplification (a) and deletion (b) in the chromosome regions. The green line indicates the significance threshold (q-value= 0.25).

The visualization of GISTIC results of hormone receptor-positive/human epidermal growth factor receptor-negative (HR+/HER2-) cohort. GISTIC plots indicate the G-scores (top) and q-values (bottom) with respect to amplification (a) and

deletion (**b**) in the chromosome regions. The green line indicates the significance threshold (q-value= 0.25).

Reviewer #2

Summary: The present manuscript evaluates the genetic architecture of breast cancer arising from Taiwanese patients. Whole exome sequencing of paired tumor normal specimens was performed which illustrated mutational events that are similar to those which have been reported in TCGA and other studies. Subsequent analyses of whole-genome doubling, shows that this event is most common in terminal negative breast cancer (TNBC) and is associated with chromosome instability. Using allelic balance the timing of different variants is mapped relative to the whole-genome doubling. The presence of whole-genome doubling is associated with homologous-recombination repair deficiency which is inferred by genomic scarring. Lastly, the whole-genome doubling is found to correlate with a specific indel score.

Critique: While the study has merit, there is very little in the work which is new and much of the manuscript is confirmatory. Potential unique elements are not clearly articulated, and significance specific to Taiwanese patients and the manifestation of breast cancer is not clear. A few points are provided below:

Response: We thank the reviewer for your valuable comments. We have made every effort possible to address your concerns. We believe that the modification and additional analyses in response to your suggestions considerably improve the manuscript. Thank you for your time and thoughtful comments on this manuscript.

1. The number of cases is relatively limited for sub-type specific interrogation. There are only 14 TNBC patients in the cohort, so sub-stratifying further limits any statistical power particularly when correcting for other variables (e.g. grade)

Response: We appreciate your valuable comment. Except for subtypes, we did not perform any further sub-stratifying for other clinical factors. In keeping with this important recommendation, we have added the bar plot with 95% CI (Bootstrapping) in the Result section 2 and new Fig. 1b to statistically confirm the differences of mutation frequencies between the BCTW and TCGA cohorts (**Change #30-34 / line 144-146, 153, 682, and 687-692**). In the Result section 3, we have also toned down the description of WGD-related alterations to “relatively frequent alterations of cancer

genes found in the WGD samples” (Change #20-27 / line 97, 169, 171-174, 194-196, 251-252, 266-267, 273-274, 509, 513, and 514). As the analysis of *WRN* might have insufficient evidence to support the associations between WGD and HRD, we have deleted the HR-related alteration analysis in the Result section 5 and Fig. 4d (Change #7-15 / line 58-59, 100-102, 215, 225, 288-290, 292, 314-316, 509, 721, and 730). Based on your helpful suggestion of point 4, in the Results section 6, we have added a "control" test to rule out the possibility of a higher contribution of de novo ID2 caused by disproportionate numbers of indels in the WGD samples. It might provide additional support for the conclusion (Change #16-19 / line 59-60, 240-244, 302-306, and 424-431).

2. Whole genome duplication and increased chromosome instability is already known to be over-represented in TNBC relative to other subtypes of breast cancer. Similarly the mutational signatures are largely consistent with what is known for sporadic breast cancer.

Response: We appreciate your valuable comments. Previous WGD studies primarily focused on one BC subtype (TNBC in López et al., 2020; HR+/HER- in Bielski et al., 2018), without comparison between BC subtypes. In the original manuscript, we had performed WGD and CIN comparisons across four subtypes. In the Abstract, we have further modified sentences to highlight the results of the other subtypes. (Change #39 / line 51-58). To our knowledge, WGD and CIN across subtypes haven't been quantitatively checked in Asian cohorts before. Therefore, we tried to examine and report relevant results of WGD across subtypes. We have added sentences to highlight this aim in the Introduction and Discussion (Change #40 and #41 / line 83 and 311-314). We have also toned down sentences in the Discussion (Change #48 / line 309-311). In the Results section 2 of the previous manuscript, we had identified the SBS signatures to bring out and focus on the changes of signature contribution from pre-WGD to post-WGD for the Results section 4.

3. The clonal assessment of the mutation events is interesting, but given the small numbers of events it would seem to be impossible to make a rigorous conclusion from pre vs. post whole genome doubling. In general the presentation would benefit by clearly disclosing the number of cases being analyzed and how they are being pooled/interrogated. Presenting individual cases may better illustrate the

point being indicated.

Response: We thank the reviewer for your valuable comments. As pointed out, generating the hypothesis for the observations of WGD is more proper than making rigorous conclusion. We have made this point in the Introduction and Discussion sections (**Change #4 and #5 / line 76-78 and 248-250**). According to your thoughtful suggestion, we have provided the number of cases in these analyses (**Change #43-47, and #49 / line 702, 707-710, 712, 727-730, 742, and 743**; 116 BCTW samples in the Results sections 2 and 6; 106 BCTW sample, which had subtype information, in the Results sections 3 and 5; and 22 BCTW samples with in the Results section 4). Thank you for your point that presenting individual cases would illustrate the individual-level clonal evolution. However, in our study, we tended to investigate the clonal assessment at the subtype level, so we presented this aim in Fig. 3.

4. The analyses of homologous repair deficiency and INDEL signatures is similarly interesting, but increased homologous repair deficiency is already known to occur in TNBC and the powering relative to the conclusion (ie. number of tumors) needs to be disclosed. Boot strapping with other data or some other approach may be necessary to provide additional support for these conclusions. This is particularly relevant with regard to the analysis of WRN.

Response: We thank the reviewer for your thoughtful comments. In the Results section 5, as the analysis of *WRN* might have insufficient evidence to support the associations between WGD and HRD, based on the reviewer 1's suggestion, we have deleted those sentences about the HRD scores increased by HR-related alterations in the manuscript and focused on the evidence from indel signature based on reviewer's thoughtful suggestion (**Change #7-15 / line 58-59, 100-102, 215, 225, 288-290, 292, 314-316, 509, 721, and 730**). In keeping with this important recommendation, we have assessed the contribution of the identified signature by a "control" test between the original WGD and "pseudo" non-WGD samples and found no significant difference. It would rule out the possibility that the higher contribution of de novo ID2 could originate from the disproportionate numbers of indels in the WGD samples, supporting our finding (**Change #16-19 / line 59-60, 240-244, 302-306, and 424-431**). We have also provided information about the number of tumors (**Change #45 / line 727-730**).

5. Some form of functional studies to support the associations in the study would be important to solidify the observations with reference to homologous repair deficiency and whole genome doubling as others have performed using different modeling systems.

Response: We appreciate your valuable comments. Although functional studies of the associations are beyond the scope of the present study, we do understand that these issues are particularly important. The suitable cell lines may need to match several conditions, such as the HRD phenotype, diploid and polyploid subclones in four subtypes. Therefore, in the Discussion, we have added sentences that future studies with suitable cell lines would support our hypothesis (**Change #42 / line 316-320**). These suitable cell lines, which contain both diploid and polyploid subclones with the HRD phenotype from human breast cancers across four subtypes, would be isolated at different passages and sequenced to validate the finding in this study.

List of Changes

Change #1 / line 515-517

We have added the BAF plot as Supplementary Figures for readers to assess the WGD analysis.

Revised, Supplementary Figures 5-26

Revised, page 21, Methods, Statistics and reproducibility

The allelic ratio plots and sequencing quality metric for assessing WGD analysis are shown in Supplementary Figures 5-26 and Supplementary Data 2.

Change #2 / line 515-517

We have also added the quality metric for readers to check the sequenced quality in the Supplementary Tables.

Revised, Supplementary Data 2

Revised, page 21, Methods, Statistics and reproducibility

The allelic ratio plots and sequencing quality metric for assessing WGD analysis are shown in Supplementary Figures 5-26 and Supplementary Data 2.

Change #3 / line 262-266

We have added descriptions in the Discussion to highlight the assumption of the deletion history.

Revised, page 11, second paragraph of Discussion

However, the underlying history of deletions is complex and uncertain, hence not identifiable except with additional strong assumptions about the CN paths that are allowable. Therefore, we assume that the current copy number paths from sequencing data can be used to uncover and determine the timing of mutation and copy number alterations, based on a prior study²³.

Change #4 / line 76-78

We have added sentences to generate the hypothesis for the observations of WGD in the Introduction.

Revised, page 4, second paragraph of Introduction

In this study, the previous biological finding of BC subtypes caused by different profiles of SNVs and CNAs² allows us to hypothesize that large-scale somatic events might also explain the subtype heterogeneity.

Change #5 / line 248-250

We have added sentences of hypothesis for the observations of WGD in the Discussion.

Original, page 11, first paragraph of Discussion

In this study, the prevalence of WGD displayed remarkable variability among subtypes, with its effects mostly observed in TNBC tumors.

Revised, page 10-11, first paragraph of Discussion

In this study, it can be hypothesized that WGD might explain distinct genomic complexity across breast cancer subtypes. The prevalence of WGD displayed remarkable variability among subtypes, with its effects mostly observed in TNBC tumors.

Change #6 / line 92-94

In the Introduction, we have added the sentence to highlight the aim of quantitative assessment for the extent of HRD in the WGD samples.

Revised, page 4, third paragraph of Introduction

Furthermore, the extent of homologous recombination deficiency in the WGD samples hasn't been quantitatively checked in the Asian cohort before.

Change #7 / line 58-59

We have deleted the sentence, which describe the enrichment of LOF in HR-related genes to support the interplay of WGD and HRD, in the Abstract.

Original, page 3, Abstract

Bi-allelic inactivation of the WRN gene, explaining the interplay of WGD and loss of HR DNA repair in triple-negative BC, and association of higher activity of *de novo* indel signature 2 with WGD and HR deficiency in Taiwanese BC patients, was further reported.

Revised, page 3, Abstract

An association of higher activity of *de novo* indel signature 2 with WGD and HRD in Taiwanese BC patients was reported.

Change #8 / line 100-102

We have modified the sentence, which describe the enrichment of LOF in HR-related genes to support the interplay of WGD and HRD, in the Introduction.

Original, page 5, last paragraph of Introduction

WGD and HRD were revealed to co-occur with bi-allelic LOF in HR-related genes,

where further genomic characterization of alternative HR repair processes confirmed indel signatures to be a better and more reliable predictor of WGD-linked HRD phenotype than the substitution signatures.

Revised, page 5, last paragraph of Introduction

WGD and HRD were revealed to co-occur with the specific indel pattern, where further genomic characterization of alternative HR repair processes revealed indel signatures to be a better and more reliable predictor of WGD-linked HRD phenotype than the substitution signatures.

Change #9 / line 215

In the Results section 5, we have toned down the sentence, which would aim to quantitatively evaluate the extent of HRD in our WGD samples.

Original, page 9-10, Results, WGD with HRD

To evaluate the biological impact of the association between WGD and HRD, the HRD score was first calculated by combining 3 genomic scar scores²⁰.

Revised, page 9, Results, WGD with HRD

To quantitatively evaluate the extent of HRD in the WGD samples, the HRD score was first calculated by combining 3 genomic scar scores²⁰.

Change #10 / line 225

We have deleted those sentences, which describe the enrichment of LOF in HR-related genes to support the interplay of WGD and HRD, in the Results section.

Original, page 10, Results, WGD with HRD

Next, the presence and timing of bi-allelic LOF alterations affecting HR-related genes¹⁰ was investigated. A total of 14.2% of tumors harbored at least one bi-allelic inactivation of HR genes, consistent with the TCGA study (10%)¹⁰. Of the 5 tumors with both WGD and HRD phenotypes affected by somatic bi-allelic loss (LOF) of HR-related genes, 4 were TNBCs (Fig. 4d), among which were predominantly pre-WGD clonal alterations (such as WGD-related deletion of *WRN* and *DNMT3A* deletion); followed by the subclonal alterations, including *FANI* and *ATM* (deletion), that occurred post-WGD. In the HR+/HER2+ tumor, a putatively somatic pathogenic mutation in exon 17 of *RPA1* (p.V597M in Rep_fac-A_C domain) with LOH was found, classified as clonal pre-WGD.

Revised, page 10, Results, WGD with HRD

Deleted.

Change #11/ line 288-290

In the Discussion, we have added and toned down sentences, which would aim to quantitatively evaluate the extent of HRD in our WGD samples.

Original, page 13, third paragraph of Discussion

Our observation confirmed that WGD-positive tumors significantly associated with HRD for all subtypes, and thus all might benefit from HRD-related treatment.

Revised, page 12, third paragraph of Discussion

We quantitatively evaluated the extent of HRD in the WGD samples. Our observation showed that WGD-positive tumors were significantly associated with HRD for all subtypes, and thus all might benefit from HRD-related treatment.

Change #12 / line 292

We have deleted those sentences, which describe the enrichment of LOF in HR-related genes to support the interplay of WGD and HRD, in the Discussion.

Original, page 13, third paragraph of Discussion

We found that a complete LOF (deep deletion) of *WRN* might contribute to both the loss of competent HR DNA repair and the presence of the WGD event in TNBC. However, it is also possible that TNBC tumors with increased CIN following WGD may have more chance to harbor bi-allelic LOF alterations or some unidentified mechanism to induce HRD.

Revised, page 12, third paragraph of Discussion

Deleted.

Change #13 / line 314-316

We have modified those sentences, which describe the enrichment of LOF in HR-related genes to support the interplay of WGD and HRD, in the Discussion.

Original, page 14, last paragraph of Discussion

Our work has outlined the subtype specificity of WGD and CIN in BCTW patients and provided insights into the genetic basis of HR-related alterations linked to WGD in BC.

Revised, page 13, last paragraph of Discussion

Our work has outlined the subtype specificity of WGD and CIN in BCTW patients and provided insights into the genomic basis of de novo ID2 linked to WGD and HRD in BC.

Change #14 / line 509

We have deleted the method about assessment of bi-allelic LOF alterations.

Original, page 21-22, Methods, Assessment of bi-allelic LOF alterations in HR-related genes

A published list of 102 core and related HR genes was assessed as previously described¹⁰. We defined LOF mutations as those with clearly functional impact on a gene, including frame shift, nonsense, start/stop codon changes, and splice site mutations, whereas missense mutations were considered as variants of uncertain/unknown significance (VUS). Germline variant calling was conducted using HaplotypeCaller with the gvcf mode from GATK v.3.740. Bi-allelic alterations in HR genes refers to cases where both alleles of an HR-related gene were lost. This included (1) a germline LOF mutation with somatic LOH of the wild-type allele, (2) a germline LOF mutation with a somatic LOF mutation in the same gene, (3) a somatic LOF mutation with somatic LOH of the wild-type allele, (4) two somatic LOF mutations in the same gene, and (5) a somatic deep deletion. On the contrary, we considered (1) a germline LOF mutation with a somatic VUS, (2) a somatic LOF mutation with a somatic VUS, or (3) a somatic VUS with somatic LOH of the wild-type allele to be a bi-allelic VUS event. Bi-allelic VUSs in HRD cases without other bi-allelic LOF alterations were considered putatively LOF events.

Revised, page 21, Methods

Deleted.

Change #15 / line 721 and 730

We have deleted the subfigure with its legend about assessment of bi-allelic LOF alterations in Fig.4.

Original, page 31 and 36, Fig. 4 Homologous recombination deficiency (HRD) and timing of bi-allelic loss-of-function (LOF) affecting HR-related genes in whole-genome doubling (WGD) cancers.

d Oncoprint indicates the incidence and timing of bi-allelic LOF of HR-related genes in 5 patients harboring both HRD and WGD. Subtypes are annotated for each tumor (top). The timing of bi-allelic LOF events is annotated by colors indicating whether the events are clonal or subclonal. Clonal bi-allelic alterations are further designated as early or late with respect to WGD.

Revised, page 33, Fig. 4 Homologous recombination deficiency (HRD) in whole-

genome doubling (WGD) cancers.

Deleted.

Change #16 / line 59-60

In the Abstract, we have added sentences about the "control" test to support the finding of a higher contribution of *de novo* ID2 associated with WGD and HRD.

Revised, page 3, Abstract

A "control" test between WGD and "pseudo" non-WGD samples was further employed to support this finding.

Change #17 / line 240-244

In the Results, we have added sentences about the "control" test to rule out the possibility of a higher contribution of *de novo* ID2 caused by disproportionate numbers of indels in the WGD samples.

Revised, page 10, Results, WGD with alternative DSB repair processes

We further performed a "control" test to rule out the possibility that a higher contribution of *de novo* ID2 is caused by disproportionate numbers of indels in the WGD samples. Our results displayed no statistical significance in 978 out of 1000 times implying that the high proportion of indels in the WGD samples had no role to play in the contribution of *de novo* ID2 in WGD.

Change #18 / line 302-306

In the Discussion, we have added sentences about the "control" test that we had ruled out the possibility of a higher contribution of *de novo* ID2 caused by disproportionate numbers of indels in the WGD samples.

Original, page 13, fourth paragraph of Discussion

It was further confirmed that signature ID2 is a significant predictor of the interplay between HRD and WGD.

Revised, page 13, fourth paragraph of Discussion

However, the higher contribution of *de novo* ID2 could originate from the disproportionate numbers of indels in the WGD samples. To rule out this possibility, a "control" test between WGD and "pseudo" non-WGD samples was employed and supported our finding. It was further observed that signature ID2 is a significant predictor of the interplay between HRD and WGD.

Change #19 / line 424-431

In the Methods, we have added a subsection about the "control" test, which was performed to rule out the possibility of enriched contribution caused by disproportionate numbers of indels in the WGD samples. We have also provided the number of indels for each sample in the Supplementary Data.

Revised, Supplementary Data 1**Revised, page 17-18, Methods, Mutational signature analysis**

To rule out the possibility of the higher contribution of *de novo* ID2 caused by disproportionate numbers of indels that are called from the WGD samples (Supplementary Data 1), a "control" test was performed. From each WGD sample, we randomly sampled indels consistent with the median indel count of the non-WGD cohort, treating it as a "pseudo" non-WGD sample, and estimated the contribution of *de novo* ID2. We subsequently checked if these control samples showed a significantly lower contribution to *de novo* ID2 than the WGD samples using the one-tailed Mann-Whitney U test. The above computations were further repeated 1,000 times to compute the probability.

Change #20 / line 97

To tone down the finding of WGD-related alterations, we have modified the sentence in the Introduction.

Original, page 4-5, last paragraph of Introduction

To address such lack of information, we performed a comprehensive analysis of deep whole-exome sequencing (WES) data from BC patients in the Taiwanese population (BCTW), to infer the impact of WGD, along with the statistical significance of WGD-related alterations, on the timing of events driving tumor initiation and tumor maintenance within subtypes of BC (hormone receptor-positive and human epidermal growth factor 2 receptor-negative (HR+/HER2-), HR+/HER2+, HER2+, and triple-negative (TNBC)).

Revised, page 4-5, last paragraph of Introduction

To address such lack of information, we performed a comprehensive analysis of deep whole-exome sequencing (WES) data from BC patients in the Taiwanese population (BCTW), to infer the impact of WGD, along with frequent cancer gene alterations, on the timing of events driving tumor initiation and tumor maintenance within subtypes of BC (hormone receptor-positive and human epidermal growth factor 2 receptor-negative (HR+/HER2-), HR+/HER2+, HER2+, and triple-negative (TNBC)).

Change #21 / line 169

To tone down the finding of WGD-related alterations, we have modified sentences in Results.

Original, page 8, Results, WGD and CIN in BC subtypes

Given the rate and variability of WGD in BCTW samples, associations between specific genetic lesions and WGD was further explored.

Revised, page 7, Results, WGD and CIN in BC subtypes

Deleted.

Change #22 / line 171-174

To tone down the finding of WGD-related alterations, we have modified sentences in the Results.

Original, page 8, Results, WGD and CIN in BC subtypes

Next, the association between WGD and recurrent alterations was determined, where genes *MYC*, *EIF4EBP1*, and *FGFR1* displayed amplifications; *DUSP4*, *LEPROTL1*, *NRG1*, and *WRN*, deletions; and *MUC16* demonstrated mutations ($P < 0.05$, odds ratio > 1 ; logistic regression). Interestingly, taken together, Gene ontology term enrichment analysis suggests that WGD-positive BCTW samples might be affected by translation initiation factor binding.

Revised, page 7, Results, WGD and CIN in BC subtypes

Next, compared to non-WGD samples, the relatively frequent alterations of cancer genes were found in WGD samples, where genes *MYC*, *EIF4EBP1*, and *FGFR1* displayed amplifications; *DUSP4*, *LEPROTL1*, *NRG1*, and *WRN*, deletions; and *MUC16* demonstrated mutations ($P < 0.05$, odds ratio > 1 ; logistic regression).

Change #23 / line 194-196

To tone down the finding of WGD-related alterations, we have modified the sentence in Results.

Original, page 9, Results, Clonality and timing of driver events within subtypes

Notably, most WGD-related genes within this subtype were primarily subclonal and occurred late.

Revised, page 8, Results, Clonality and timing of driver events within subtypes

Notably, deletion of *DUSP4*, *LEPROTL1*, *NRG1*, and *WRN* within this subtype were primarily subclonal and occurred late.

Change #24 / line 251-252

To tone down the finding of WGD-related alterations, we have modified sentences in the Discussion.

Original, page 11, first paragraph of Discussion

Notably, 8 somatic alterations exhibited a statistically significant association with WGD across BC subtypes. It might be inferred that dysfunction of translation initiation factor binding increases the likelihood of a tumor undergoing WGD in BCTWs.

Revised, page 11, first paragraph of Discussion

Moreover, eight somatic alterations frequently altered in the WGD samples were found across BC subtypes.

Change #25 / line 266-267

To tone down the finding of WGD-related alterations, we have modified the sentence in the Discussion.

Original, page 12, second paragraph of Discussion

FGFR1 amplification, an early clonal WGD-related event in HR+ tumors, conferred antiestrogen resistance to ER+ BC²⁴.

Revised, page 11, second paragraph of Discussion

FGFR1 amplification, an early clonal somatic event in HR+ tumors, conferred antiestrogen resistance to ER+ BC²⁴.

Change #26 / line 273-274

To tone down the finding of WGD-related alterations, we have modified the sentence in the Discussion.

Original, page 12, second paragraph of Discussion

In TNBC, WGD-related deletions occurred before WGD, in contrast to HR+/HER2+ tumors, in which they were post-WGD.

Revised, page 11, second paragraph of Discussion

In TNBC, deletions of *DUSP4*, *LEPROTL1*, *NRG1*, and *WRN* occurred before WGD, in contrast to HR+/HER2+ tumors, in which they were post-WGD.

Change #27 / line 509, 513, and 514

To tone down the finding of WGD-related alterations, we have modified the sentence in the Methods.

Original, page 22, Statistical analysis

Logistic regression was performed to identify WGD-related genes.

Revised, page 21, Statistics and reproducibility

Logistic regression was performed to identify relatively frequent cancer gene alterations in the WGD samples.

Change #28 / line 58

To tone down the finding of *MYC*, we have deleted the sentence in the Abstract.

Original, page 3, Abstract

WGD-associated somatic alterations, including *MYC* amplification, were identified across BC subtypes.

Revised, page 3, Abstract

Deleted.

Change #29 / line 277

To tone down the finding of *MYC*, we have deleted the sentence in Discussion.

Original, page 12, second paragraph of Discussion

Also, except in HR+/HER2+ and HER2+ tumors, *MYC* amplification promoted early tumorigenesis by inducing proliferation, inhibiting exit from the cell cycle, stimulating vascularization, and enhancing genomic instability^{26,27}, as did mutation of *TP53*. Taken together, targeting *MYC* amplification might provide a means of maintenance of genome integrity and early therapeutic intervention.

Revised, page 12, second paragraph of Discussion

Deleted.

Change #30 / line 682 and 687-692

We have added bar plot with 95 % CIs in Fig. 1b and modified the figure legend of Fig. 1a.

Original, page 29, 30, and 33, Fig. 1 Mutational landscape of somatic alteration and single base substitution mutational signatures (SBSs) in 116 breast cancer samples

a Rows represent significantly mutated genes (SMGs), and columns represent individual tumors. Samples are arranged to emphasize mutual exclusivity among alterations. SMGs are ordered according to the frequency of nonsynonymous single nucleotide variations/indels, compared with those from The Cancer Genome Atlas

benchmark cohorts (right; TW, Taiwan; CCSN, Caucasian; AFRAM, African American; ASAM, Asian American). The stacked bar plot (top) depicts the tumor mutation burden (TMB; mutations/covered bases; y-axis) for individual tumors (x-axis). Key clinical features are annotated for each tumor (bottom). Clinical characteristics and mutation types are indicated with color.

Revised, page 29 and 30, Fig. 1 Mutational landscape of somatic alteration and single base substitution mutational signatures (SBSs) in 116 breast cancer samples

a Rows represent significantly mutated genes (SMGs), and columns represent individual tumors. Samples are arranged to emphasize mutual exclusivity among alterations. SMGs are ordered according to the frequency of nonsynonymous single nucleotide variations/indels. The stacked bar plot (top) depicts the tumor mutation burden (TMB; mutations/covered bases; y-axis) for individual tumors (x-axis). Key clinical features are annotated for each tumor (bottom). Clinical characteristics and mutation types are indicated with color. **b** The bar plot with the 95% confidence interval indicates mutational frequencies of BCTWs, compared with those from The Cancer Genome Atlas benchmark cohorts (BCTW, Taiwanese; CCSN, Caucasian; AFRAM, African American; ASAM, Asian American).

Change #31 / line 692

As we added a new Fig. 1b, the original Fig. 1b was renumbered as new Fig. 1c.

Original, page 29 and 30, Fig. 1 Mutational landscape of somatic alteration and single base substitution mutational signatures (SBSs) in 116 breast cancer samples

b Heatmap of the cosine similarity results for the 3 *de novo* SBSs of the Taiwanese population (y-axis), coded by color.

Revised, page 30, Fig. 1 Mutational landscape of somatic alteration and single base substitution mutational signatures (SBSs) in 116 breast cancer samples

c Heatmap of the cosine similarity results for the 3 *de novo* SBSs of the Taiwanese population (y-axis), coded by color.

Change #32 / line 144

In the Results, we have modified the sentence for the new Fig. 1b.

Original, page 6, Results, Mutational landscape of BCTW samples.

Comparative analysis across populations was briefly performed using BCTWs and The Cancer Genome Atlas (TCGA) benchmark cohorts (Fig. 1a, right)

Revised, page 6, Results, Mutational landscape of BCTW samples.

Comparative analysis across populations was briefly performed using BCTWs and The Cancer Genome Atlas (TCGA) benchmark cohorts (Fig. 1b)

Change #33 / line 153

In the Results, we have modified a sentence for the new Fig. 1b.

Original, page 7, Results, Mutational landscape of BCTW samples.

Our analysis also identified somatic mutations in genes that turned out to be only locally prevalent (*ERN1*, *PIGT*, and *COMP*) (Fig. 1a, right).

Revised, page 7, Results, Mutational landscape of BCTW samples.

Our analysis also identified somatic mutations in genes that turned out to be only locally prevalent (*ERN1*, *PIGT*, and *COMP*) (Fig. 1b).

Change #34 / line 145 and 146

In the Results, we have modified sentences based on the bar plot with 95 % CIs in new Fig. 1b.

Original, page 7, Results, Mutational landscape of BCTW samples.

Genes *ARID1A*, *PTEN*, *KMT2A*, and *RBI* displayed similar mutation prevalence for both BCTW and TCGA cohorts.

Revised, page 6, Results, Mutational landscape of BCTW samples.

Genes *ARID1A*, *KMT2A*, and *RBI* displayed similar mutation prevalence for both BCTW and TCGA cohorts, except for Caucasians.

Change #35 / line 682, 693, and 694

To avoid confusion, we have added the detailed information in new Fig. 1c and modified the figure legend.

Original, page 29, 30, and 33, Fig. 1 Mutational landscape of somatic alteration and single base substitution mutational signatures (SBSs) in 116 breast cancer samples

b Heatmap of the cosine similarity results for the 3 *de novo* SBSs of the Taiwanese population (y-axis), coded by color. The cosine similarity (range 0-1) represents the extent of similarity to a particular signature of COSMIC (x-axis). Among the 30 COSMIC SBSs, the APOBEC- and age-related signatures were the most similar mutational signatures detected in the Taiwanese population (dark red), while one signature was dissimilar to any COSMIC mutational signatures and thus is considered unknown.

Revised, page 29 and 30, Fig. 1 Mutational landscape of somatic alteration and single base substitution mutational signatures (SBSs) in 116 breast cancer samples

c Heatmap of the cosine similarity results for the 3 *de novo* SBSs of the Taiwanese population (y-axis), coded by color. In the scale bar, the cosine similarity (range 0-1) represents the extent of similarity to a particular signature of COSMIC (x-axis). Among the 30 COSMIC SBSs, the APOBEC- and age-related signatures were the most similar mutational signatures detected in the Taiwanese population (dark red), while one signature was dissimilar to any COSMIC mutational signatures and thus is considered unknown.

Change #36 / line 731 and 734

To avoid confusion, we have added the detailed information in Fig. 5a and modified the figure legend.

Original, page 31, 32, and 37, Fig. 5 Indel mutational signatures (IDs) with homologous recombination deficiency (HRD) and whole-genome doubling (WGD)

a Heatmap of the cosine similarity results for the *de novo* IDs (y-axis), coded by color. The cosine similarity (range 0-1) represents the extent of similarity to a particular signature of COSMIC (x-axis).

Revised, page 34 and 35, Fig. 5 Indel mutational signatures (IDs) with homologous recombination deficiency (HRD) and whole-genome doubling (WGD)

a Heatmap of the cosine similarity results for the *de novo* IDs (y-axis), coded by color. In the scale bar, the cosine similarity (range 0-1) represents the extent of similarity to a particular signature of COSMIC (x-axis).

Change #37 / line 700

We have corrected the issue of the discrepancy between Figure 2c and line 184 in new Fig. 2c.

Revised, page 31, Fig. 2 Prevalence of whole-genome doubling (WGD), level of chromosomal instability (CIN), and timing of WGD across breast cancer (BC) subtypes.

Change #38 / line 711, 716, 717, and 720

We have labeled amplifications and deletions, corrected the coloring and modified the figure legend in Fig. 3 and Supplementary Figures 2-4.

Original, page 31 and 35 , Fig. 3 Timing of somatic events in triple-negative breast

cancer (TNBC) and Supplementary Figures 2-4

b The timing of mutations and copy number events is shown as bars indicating whether the events are clonal or subclonal. Clonal mutations and chromosome-arm events are further designated as early or late with respect to whole-genome doubling (WGD). The frequency of mutations and copy number alterations (CNAs, pre-GD and post-GD) is indicated on the right side of the bars. **c** Pie charts show the percentage of mutations for each signature, averaged across the TNBC cohort. Only genes that were mutated in Cancer Gene Census or canonical signaling pathways in the cohort are shown.

Revised, page 32, Fig. 3 Timing of somatic events in triple-negative breast cancer (TNBC) and Supplementary Figures 2-4

b The timing of mutations and copy number events is shown as bars indicating whether the events are clonal or subclonal. Clonal mutations and chromosome-arm events are further designated as early or late with respect to whole-genome doubling (WGD). The number of samples harboring mutations and copy number alterations (CNAs, pre-GD and post-GD) is indicated on the right side of the bars. **c** Pie charts show the percentage of mutations for each signature, averaged across the TNBC cohort. Only genes that were mutated in Cancer Gene Census or canonical signaling pathways in the cohort are shown. AMP: amplification; DEL: deletion.

Change #39 / line 51-58

In the original manuscript, we had performed WGD and CIN comparisons across four subtypes. In the Abstract, we have further modified sentences to highlight the results of the other three subtypes.

Original, page 3, Abstract

Whole-genome doubling (WGD) is an early macro-evolutionary event in tumorigenesis, involving the doubling of an entire chromosome complement. However, its impact on breast cancer (BC) subtypes has not been explored before. Here, we performed a comprehensive analysis of WGD and its influence on BC subtypes in patients from Taiwan and consequently highlights the genomic association between WGD and homologous recombination (HR) deficiency. A higher manifestation of WGD was reported in triple-negative BC, conferring high chromosomal instability.

Revised, page 3, Abstract

Whole-genome doubling (WGD) is an early macro-evolutionary event in tumorigenesis, involving the doubling of an entire chromosome complement. However,

its impact on breast cancer (BC) subtypes remains unclear. Here, we performed a comprehensive and quantitative analysis of WGD and its influence on BC subtypes in patients from Taiwan and consequently highlight the genomic association between WGD and homologous recombination deficiency (HRD). A higher manifestation of WGD was reported in triple-negative BC, conferring high chromosomal instability (CIN), while HER2+ tumors exhibited early WGD events, with widely varied CIN levels, compared to luminal-type tumors.

Change #40 / line 83

We have added a sentence to highlight the aim of WGD analyses for the Asian cohort in the Introduction.

Original, page 4, second paragraph of Introduction

However, the extent of WGD and CIN within subtypes of BC has not been explored yet.

Revised, page 4, second paragraph of Introduction

However, the extent of WGD and CIN within subtypes of BC has not been explored yet in the Asian cohort.

Change #41 / line 311-314

We have added a sentence to highlight our aim of WGD analyses in the Discussion.

Original, page 14, last paragraph of Discussion

The impact of WGD and the timing of genome abnormalities across subtypes of BC, which have not been characterized before, are explored in this study.

Revised, page 13, last paragraph of Discussion

The impact of WGD and the timing of genome abnormalities across subtypes of BC, which have not been characterized in the Asian cohort before, are explored in this study. In the BCTW cohort, we observed the subtype heterogeneity of WGD, which can be investigated whether the same situation occurs in other populations. We expect that more population studies using comprehensive sequencing approaches for WGD will support our finding.

Change #42 / line 316-320

In the Discussion, we have added sentences that future studies with suitable cell lines would support our hypothesis

Revised, page 13, last paragraph of Discussion

Future studies with suitable cell lines, which contain both diploid and polyploid subclones with HRD phenotype from human BCs in four subtypes, would be needed to support our hypothesis. These suitable cell lines would be isolated at different passages and sequenced to validate the findings in this study. However, future cell line studies would be out of the scope of the current study.

Change #43 / line 702 and 707-710

In Fig. 2, we have added number of cases in the figure legend.

Original, page 30, Fig. 2 Prevalence of whole-genome doubling (WGD), level of chromosomal instability (CIN), and timing of WGD across BC subtypes

c Box plots indicate the timing of WGD in each subtype. Timing was estimated based on the fraction of mutations occurring post-WGD compared to pre- and post-WGD combined. HR, hormone receptor; HER2, human epidermal growth factor 2 receptor; TNBC, triple-negative breast cancer.

Revised, page 31, Fig. 2 Prevalence of whole-genome doubling (WGD), level of chromosomal instability (CIN), and timing of WGD across breast cancer subtypes

c Box plots indicate the timing of WGD in each subtype. Timing was estimated based on the fraction of mutations occurring post-WGD compared to pre- and post-WGD combined. Overall, 64 HR+/HER2- (hormone receptor-positive (HR+)/human epidermal growth factor receptor-negative), 13 HR+/HER2+, 15 HER2+, and 14 TNBC (triple-negative breast cancer) samples were used to perform analyses in a-c.

Change #44 / line 712

In Fig. 3, we have added number of cases in the figure legend.

Original, page 30, Fig. 3 Timing of somatic events in triple-negative breast cancer (TNBC).

Revised, page 32, Fig. 3 Timing of somatic events in triple-negative breast cancer (TNBC; N=7).

Change #45 / line 727-730

In Fig. 4, we have added number of cases in the figure legend.

Revised, page 33, Fig. 4 Homologous recombination deficiency (HRD) in whole-genome doubling (WGD) cancers

Overall, 64 HR+/HER2- (hormone receptor-positive (HR+)/human epidermal growth factor receptor-negative), 13 HR+/HER2+, 15 HER2+, and 14 TNBC (triple-negative

breast cancer) samples were used to perform analyses in a-c.

Change #46 / line 742 and 743

In Fig. 5, we have added number of cases in the figure legend.

Revised, page 35, Fig. 5 Indel mutational signatures (IDs) with homologous recombination deficiency (HRD) and whole-genome doubling (WGD)

Overall, 116 breast cancer samples were used to perform analyses in a-d.

Change #47

In the Supplementary Figures 2-4, we have added number of cases in the figure legend.

Revised, Supplementary Figures 2-4

Change #48 / line 309-311

We have toned down sentences in the Discussion.

Original, page 14, last paragraph of Discussion

This study, to our knowledge, is the first of its kind to hypothesize that WGD is associated with specific BC subtypes, and highlights the association between HRD and WGD through an extensive analysis using WES data from BCTW patients.

Revised, page 13, last paragraph of Discussion

In this study, we found that WGD might exhibit distinct genomic complexity across BC subtypes, and highlights the association between HRD and WGD through an extensive analysis using WES data from BCTW patients.

Change #49 / line 515

We have added a sentence about sample sizes in the Methods.

Revised, page , Methods, Statistics and reproducibility

Sample sizes are included in each figure legend.

REVIEWERS' COMMENTS:

Reviewer #1 (Remarks to the Author):

The authors have addressed my concerns. The results presented in the revised manuscript raise a clearer and more focused hypothesis regarding WGD in breast cancer. I personally think the paper is suitable for publication.